



# Gap geometry, seasonality and associated losses of biomass - combining UAV imagery and field data from a Central Amazon forest

Adriana Simonetti[1,2], Raquel Fernandes Araujo[2], Carlos Henrique Souza Celes[2], Flávia Ranara da Silva e Silva[1,2], Joaquim dos Santos[2]; Niro Higuchi[2], Susan Trumbore[3] and Daniel Magnabosco Marra[3]

[1]Programa de Pós-graduação em Ciências de Florestas Tropicais, Instituto Nacional de Pesquisas da Amazônia, Manaus, 69060-062, Brazil
[2]Laboratório de Manejo Florestal, Instituto Nacional de Pesquisas da Amazônia, Manaus, 69060-062, Brazil
[3]Biogeochemical Processes Department, Max Planck Institute for Biogeochemistry, Jena, 07745, Germany

*Correspondence to*: Adriana Simonetti (adrianasimonettilp@gmail.com) and Daniel M. Marra (dmarra@bgc-jena.mpg.de)

**Abstract.** Understanding mechanisms of tree mortality and geometric patterns of canopy gaps is relevant for robust estimates of carbon stocks and balance in tropical forests, and for assessing how they are responding to climate change. We combined monthly RGB images acquired from an unmanned aerial vehicle with field surveys to identify gaps in an 18-ha permanent plot

in an old-growth Central Amazon forest over a period of 28 months. In addition to detecting, we measured the size and shape of gaps, and analyzed their temporal variation and correlation with rainfall. We further described associated modes of tree mortality or branch fall and quantified associated losses of biomass. Overall, the sensitivity of gap detection differed between field surveys and imagery data. In total, we detected 32 gaps either in the images and field, ranging in area from 9 m$^2$ to 835 m$^2$. Relatively small gaps (< 39 m$^2$) associated with branch fall were the most frequent (11 gaps). Out of 18 gaps for which

both field and imagery data were available, three could not be detected remotely. This result shows that a considerable fraction of tree mortality and branch-fall events (~ 17 %) affect only the lower canopy and the understory of the forest and thus, are likely neglected by assessments of top of the canopy. Regardless the detection method, the size distribution of gaps in our study region was better captured by a Weibull function. As confirmed by our detailed field surveys, we believe that this pattern was not biased by gaps possibly undetected from image data. Although not related to differences in gap size, the main modes

of tree mortality partially explained associated losses of biomass. The rate of gap area formation expressed as the percent per month was positively correlated with the frequency of extreme rainfall events, which may be related to a higher frequency of storms propagating destructive wind gusts. Our results demonstrate the importance of combining field observations with remote sensing methods for monitoring gap dynamics in dense forests. The correlation between modes of tree mortality and gap geometry with associated losses of biomass provide evidence on the importance of small-scale events of tree mortality

and branch fall as processes that contribute to landscape patterns of carbon balance and species diversity in Amazon forests. Regional assessments of the dynamics and geometry of canopy gaps formed from branch fall and individual tree-mortality (e.g., from few to hundreds of m$^2$) up to catastrophic blowdowns associated with extreme rain and wind (e.g., from hundreds of m$^2$ to thousands of ha) can reduce the uncertainty of landscape assessments of carbon balance, especially as the frequency and intensity of storms causing these events is likely to change with future Amazon climate.



## 1 Introduction

Tropical forests store ~25 % of terrestrial biomass carbon stocks (Pan et al., 2013). The maintenance of these stocks depends on multi-scale and -temporal processes regulating the growth and mortality of trees (Brienen et al., 2015; McDowell et al., 2018; Frelich, 2016). Reports of increased tree mortality in tropical and temperate regions raise questions about the influence of climate change on the dynamics and functioning of old-growth forests (Laurance et al., 2004; Phillips and Gentry, 1994; Allen et al., 2015). In the tropics, climate change is related to increased frequency and intensity of extreme events, such as convective storms (Gloor et al., 2013; Tan et al., 2015; IPCC, 2021) that can increase rates of tree mortality and/or branch fall, thereby altering patterns of forest biomass and carbon (Laurance et al., 2004; Chambers et al., 2013; McDowell et al., 2018). In this context, understanding mechanisms of disturbance and forest response is fundamental to improve estimates of carbon stocks and balance from the stand to the landscape scale, and to anticipate the response of forests to varying climate scenarios (Clark et al., 2017; Leitold et al., 2018).

Gaps are natural openings in the forest canopy caused by falling trees and/or branches (Brokaw, 1982; Whitmore, 1989). Such disturbances exert great influence on the dynamics and functioning of tropical forests, as they alter structure (Kellner et al., 2009), natural regeneration (Grubb, 1977; Kellner and Asner, 2014), species diversity and composition (Denslow, 1987; Magnabosco Marra et al., 2014a, 2018), soil carbon stocks and nutrients (Santos et al., 2016; Vitousek and Denslow, 1986), and productivity (Baker et al., 2004). The size of gaps can vary from a few square meters to thousands of hectares, depending on the mechanism of formation (Nelson et al., 1994; Fontes et al., 2018; Magnabosco Marra et al., 2018; Esquivel-Muelbert et al., 2020; Araujo et al., 2017, 2021). The size and shape of gaps define the amount of light and other key resources during succession (Denslow, 1980, 1987; Schliemann and Bockheim, 2011). Apart from related to mechanisms of formation, the size and shape of gaps can be influenced by local climate and topography, soil and forest structure and species composition (Denslow, 1987; Araujo et al., 2021; Cushman et al., 2022). Thus, assessing the size distribution of gaps provides information on key processes regulating forest structure and diversity, and related functions (Jucker, 2022).

Traditionally, studies of gap dynamics and geometry (e.g., area, perimeter and shape) have relied on observations made as part of forest inventories (Brokaw, 1982; Hubbell et al., 1999). However, gap-forming events can be stochastic and obtaining robust information on their frequency and geometry from often relatively low number of plots surveyed infrequently is a challenging task. In recent years, studies of gap frequency and geometry have been conducted using fine-scale remote sensing, which allows for inferences across larger spatial scales (Getzin et al., 2014; Araujo et al., 2021; Asner et al., 2013; Dalagnol et al., 2021). Variations in the sensitivity of detection and spatial scales addressed using remote-sensing methods allowed for revisiting classic definitions of gap, as well as the importance and applicability of different metrics. In the field, a gap can be defined by an opening in the forest canopy extending from the upper stratum to an average height of two meters above ground (Brokaw, 1982). Optical remote sensing allows for the monitoring of relatively larger areas at high accuracy and spatial resolution (Senf, 2022; Frolking et al., 2009). However, it is limited to the detection of disturbances in the upper canopy layer.



In the Amazon, studies using intermediate spatial-resolution remote-sensing data have shown that small gaps are more frequent than relatively larger events, such as large gaps associated with convective storms (Nelson et al., 1994; Chambers et al., 2013; Araujo et al., 2017). However, the use of these data such as Landsat (30 m x 30 m pixel, 0.09 ha) do not allow mapping the

smaller and more frequent disturbances (e.g., < 0.1 ha), including those only affecting the lower canopy of the forest. As demonstrated for the region of Manaus (Brazil), Landsat images are only sensitive in detecting mortality events involving from 6 to 8 fallen trees (Negrón-Juárez et al., 2011; Chambers et al., 2013). This mismatch between the monitoring of gap dynamics using forest inventory and satellite data highlights the lack of knowledge on mechanisms of formation of relatively smaller and more frequent gaps, and thus of their influence on landscape patterns of forest dynamics and biomass balance.

An alternative to assess the full gradient of gap size and geometry is the photogrammetry computed from unmanned aerial vehicle (UAV) imagery, commonly known as drones. In addition to a more detailed description of size and geometry, successive imaging acquired with UAVs makes it possible to monitor gap dynamics at higher spatial and temporal resolutions than that provided by satellite imagery (Getzin et al., 2014; Araujo et al., 2021; Senf, 2022). Still, the monitoring of gap dynamics using high spatial and temporal resolution imagery must be validated with field data. When combined with

continuous forest inventories, UAV imagery may allow for the quantification of the relative contribution of mechanisms of gap formation such as different modes of tree mortality and branch fall, and to compute associated losses of biomass. The combination of high spatial-resolution imagery and field data also offers an unique opportunity to describe the seasonality of tree-mortality events and possible interactions with extreme weather events and their relevance for the maintenance of carbon stocks (Esquivel-Muelbert et al., 2020).

In this study, we combined high-resolution photogrammetry with detailed forest inventory data to quantify the size distribution and geometry of gaps, the relative contribution of different modes of tree-mortality and branch fall, and associated losses of biomass in an 18-ha Amazon forest. We addressed the following questions: i) How sensitive is RGB photogrammetry acquired with UAV for the detection of gaps compared with forest inventory data? ii) Is there a difference in the size distribution and geometry of gaps measured with photogrammetry and forest inventory? iii) Are gap geometry and biomass losses influenced

by traceable modes of tree mortality? iv) Is the rate and size of gap formation related to rainfall?

## 2 Methods

### 2.1 Study area

The study was conducted on a permanent plot (2°36′47″ S; 60°08′41″ W) monitored within the Project *Interação Vento-Árvore na Amazônia* (INVENTA), which is part of the Amazon Tall Tower Observatory (ATTO) (Fig. 1a). This plot (hereafter referred

to as INVENTA plot) is located at the *Estação Experimental de Silvicultura Tropical* (EEST) from the *Instituto Nacional de Pesquisas da Amazônia* (INPA), a reserve with 21,000 ha of contiguous old-growth forest (Fig. 1b). The EEST is accessible via the local road ZF-2, located at km 50 of the BR-174 highway north of Manaus, Brazil (Fig. 1c). The INVENTA plot has a size of 18 ha (600 m x 300 m) and is divided into 20 m x 20 m subplots (total of 450 subplots), which are subdivided into four



10 m x 10 m quadrats (total of 1,800 quadrats). The INVENTA plot was established in 2000 as part of the Jacaranda Project
(Pinto et al., 2003). At the time it started, all trees, palms and lianas with DBH (diameter at breast height, 1.3 m) ≥5 cm were
recorded. In 2017, prior to the start of INVENTA, all trees and palms with DBH ≥ 10 cm were remeasured.

The canopy trees in our study region are 28.65 m ± 0.46 m tall (mean ± standard deviation) (Araujo, 2019). The forest
understory and canopy are dense and closed. The richness of 10 cm DBH trees can exceed 280 species ha[-1] (Oliveira and Mori,
1999). The INVENTA plot has an undulating topography typical of the region, including areas of plateau, slope and valley.
The mean annual precipitation and temperature in the Manaus region are 2,231 ± 118 mm year[-1] (mean ± 95 % confidence
interval) and 26.9 ± 0.17 °C, respectively (1970-2016 period) (Magnabosco Marra et al., 2018). The region experiences three
consecutive months (mostly commonly from July to September) with less than 100 mm of rainfall (Negrón-Juárez et al., 2017;
Wu et al., 2016).

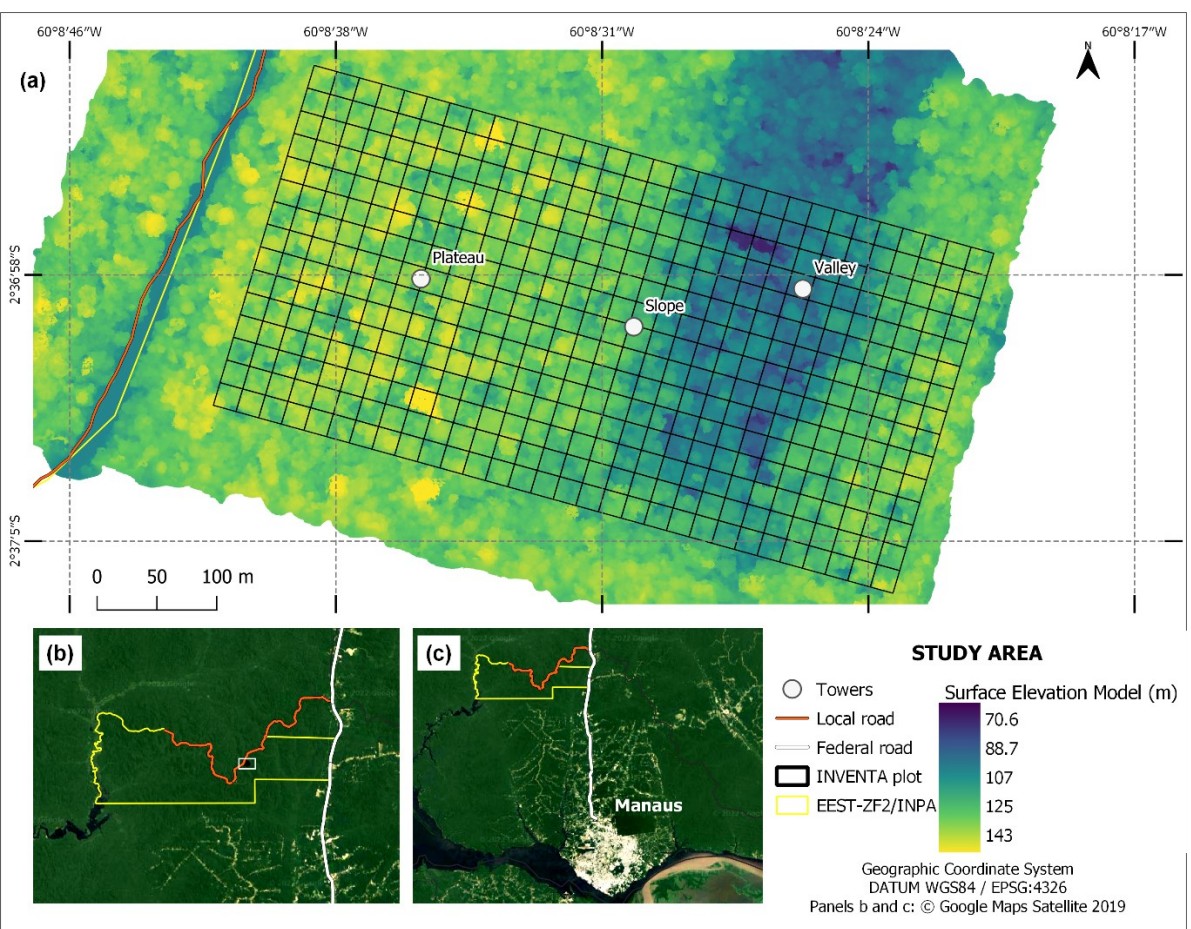

**Figure 1: Study area (INVENTA plot) with an area of 18 ha (300 m x 600 m), located ~50 km north of Manaus, Central Amazon,**
**Brazil. Elevation refers to the canopy surface model generated from photogrammetry of images obtained with UAV.**



## 2.2 Acquisition and processing of remote sensing data

Imagery data were collected monthly, between September 2018 and January 2021 (28 months), using a digital RGB camera deployed on *DJI Phantom 3* and *4* UAVs (see collection period in Table S1). The flight plans were programmed using the *DJI*
*Ground Station* application installed on a tablet device (Apple, model A1489), which was connected to the aircraft remote control and configured for automated flight from predefined waypoints. The camera lens has a Field of View (FOV) angle of 94°, and the pictures generated have a resolution of 12 Mp, with maximum dimensions of 4,000 pixels x 3,000 pixels. The overflights were performed at 100 m height above the ground, with an approximate speed of 9.9 m s$^{-1}$ in order to generate images with ~100 m width at canopy height. Photographs were captured every 2 seconds with 85 % longitudinal overlap, and
70 % lateral overlap with respect to the ground. The camera was calibrated on each flight to reduce the effects of varying illumination within and between flights. To ensure homogeneous images and diffuse lighting conditions throughout the studied period, whenever possible, flights were performed in mid-morning and/or late afternoon.

The acquired photos were processed using *Agisoft Metashape* (Version 1.5.2) (AGISOFT LLC., St. Petersburgh, Russia). This software aligns photos using the Scale Invariant Feature Transformation (SIFT) algorithm (Lowe, 2004), which allows for
ratifying photos with a bending angle greater than three degrees. Through this procedure photos were aligned from overlapping common features (i.e., textures). Further, these aligned points were given X, Y, Z coordinates and the parallax effect seen on the overlapping photos was used for reproducing the stereoscopic (3D) view based on the Structure from Motion (SfM) method. After creating the 3D point network, a dense cloud of XYZ points was generated to fill empty spaces (i.e., Dense Point Cloud). From the Dense Point Cloud, a digital surface model (DSM) and an orthomosaic were generated. The DSM is a digital
geographic dataset that represents surface elevations with horizontal and vertical (X, Y, Z) coordinates (Iglhaut et al., 2019). The orthomosaic reproduces the real dimensions of objects (Araujo et al., 2020), with horizontal spatial resolution ranging from 3 cm to 7 cm.

The orthomosaic and DSM were aligned vertically and horizontally using the georeferencing process from LiDAR data collected along transects as part of the EBA project (Ometto et al., 2021), which covered the INVENTA plot. The workflow
consisted of creating a georeferenced project based on control points extracted from LiDAR. Subsequential flights were matched using the 'Align Chunks' tool available in Agisoft Metashape (more detail on the process in Text S1).

### 2.2.1 Detection of canopy gaps

Canopy gaps within the UAV images were identified through the combination of DSM change analysis, visual interpretation of the orthomosaics (Fig. 2) and field data. Initially, we resampled the pixel resolution of photos to 1 m, and the difference-
image was calculated for all pairs to obtain a raster product (i.e., difference-image) describing changes in canopy height among time intervals.

In order to compare our data with previous studies, the area of the identified gaps was computed as the region where the canopy lost more than 10 m in height over continuous areas larger than 5 m$^2$. This was also the smallest gap size reported in previous





studies (Brokaw, 1982; Hubbell et al., 1999), with an area/perimeter ratio greater than 0.6. By computing the area/perimeter

ratio, we were able to remove artifacts associated with slight changes on the positions of individual trees in subsequential pairs

of images, both due to wind-driven canopy shifts and changes in tree alignment. Therefore, the criterion for gap identification

was based on the analysis of gap size and shape.

Finally, successive pairs of orthomosaics covering subplots (400 m$^2$) were visually checked. When necessary, we edited the

pre-delineated polygons by removing false gaps related to image noise (Araujo et al., 2021).

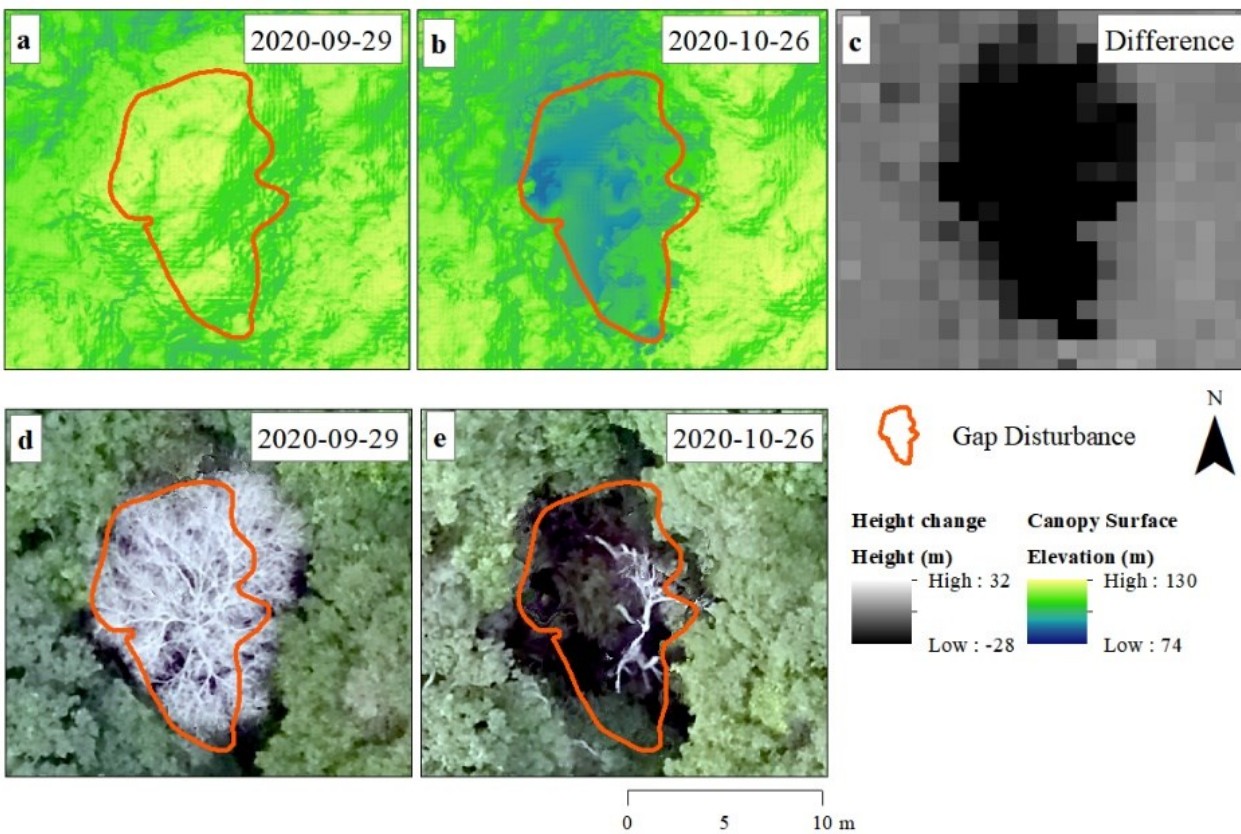

**Figure 2: Canopy gaps identified from surface models and orthomosaics computed from photogrammetric analyses of UAV imagery.**
**Elevation model for a studied gap on two successive flights from 29$^{th}$ September 2020 (a) and 26$^{th}$ October 2020 (b). The difference**
**in surface elevation between flights (black area) indicates a reduction in canopy height (c). RGB orthomosaics from the same dates**
**(d, e).**



## 2.3 Field surveys for evaluating remotely sensed gaps

Field data were collected bimonthly (section 2.3.1) and included the identification and description of gaps formed between November 2019 and January 2021 (14 months) (see collection period in Table S1). Initially, we identified all gaps formed before the studied period on the images and marked them in the field to create a reference baseline. The identification and description of gaps in the field were conducted by walking the entire plot. This task was always carried out by the same team using existing trails 10 m distant from each other to ensure precise counting and description of gaps. For field measurements, we adopted the definition of canopy gap by Brokaw (1982) - an opening in the forest canopy extending from the upper stratum to an average height of two meters above ground. In addition to confirming the gaps identified in the images, the field surveys included detailed walking of the entire plot to identify gaps possibly not detected in the images. The delimitation of gaps in the field was made by taking the coordinates (distance and azimuth) from the near center to the edge of the gap. We defined the boundaries of gaps by projecting the canopy aperture to the ground. For distance and azimuth measurements, a *TruPulse* 360B laser rangefinder (Laser Technology) was used. The center of the gap was defined in the field, and coordinates were collected by averaging Global Navigation Satellite System (GNSS) navigation points. From the center of the gap, the acquisition of eight directions and distances to the gap boundary was done counterclockwise, with the first measurement pointing north (360º/0º). The data from each gap was vectorized in QGIS Geographic Information System (version 3.4.13) (Open Source Geospatial Foundation Project. http://qgis.osgeo.org) environment from the center point. We then calculated geometric features, including gap area, perimeter, and shape complexity index.

## 2.3.1 Mechanisms of gap formation and biomass estimation

After delimiting gaps, we measured forest-structure attributes. For dead trees, the tag number, number of plot and sub-plot, diameter at breast height (DBH, 1.3 m above the ground) and the mode of mortality were recorded. We described modes of tree mortality based on previous studies conducted in our study region (Magnabosco Marra et al., 2014a; Ribeiro et al., 2016): (i) Standing dead - trees without leaves and/or presence of sap in the trunk; standing-dead trees can form or expand gaps through falling branches or the later breakage of the main stem; (ii) Snapping – trees that died from the mechanical rupture of the stem, with sap often still present at the portion connected to the roots, exposed wood fibers and no clear damaged or exposed roots; (iii) Uprooting - uprooted trees with the main trunk usually intact and still connected to the crown.

Tree biomass was estimated using a simple-entry allometric equation calibrated locally (Magnabosco Marra et al., 2016). For branches with diameter ≥5 cm, the volume was obtained by cubing combining the Smalian (measuring diameters at the base and top) and Hohenald (relative section length division) cubing methods (Lima et al., 2012; Gimenez et al., 2017). Most of the branches had no fresh vegetative material that allowed taxonomical identification to the species level. Thus, we estimated branch biomass by multiplying the measured volume of branches by the mean wood-density value compiled for our study region (0.735 [0.480,1.000], being mean wood density (g cm$^{-3}$) and minimum and maximum, respectively) (Magnabosco Marra et al., 2016).





### 2.4 Rainfall data

Rainfall data covering the studied period were acquired from a rain gauge installed at the EEST/INPA and located about 2 km from the INVENTA plot. Total daily precipitation was annotated manually. The dry season was defined as the months in which total rainfall was lower than the monthly average throughout the monitored period. For that, we used a threshold rainfall of < 200 mm (July, August, September, and October) because there were no consecutive months with rainfall ≤ 100 mm (Negrón-Juárez et al., 2017; Wu et al., 2016) (Fig. S1) during the period of this study. We also identified days with extreme rain events, which were defined as those when the accumulated precipitation was higher than the 99th percentile calculated for the entire studied period.

### 2.5 Data analysis

#### 2.5.1 Remote sensing and field detection of gaps

We used a confusion matrix to assess the accuracy of our remote method of gap detection. We calculated the percentiles of accuracy ($a$), precision ($p$), recall ($r$), and F1 Score ($F$) (Eqs. 1 − 4) (Dalagnol et al., 2021), where TP is true positive, TN is true negative, FP is false positive and FN is false negative:

$$\textit{Accuracy (a)} = ((TP+TN)\ /n)\ *100 \tag{1}$$

$$\textit{Precision (p)} = (TP/(TP+FP))\ *100 \tag{2}$$

$$\textit{Recall (r)} = (TP/(TP+FN))\ *100 \tag{3}$$

$$\textit{Score F1 (F)} = (((2*p*r)\ /\ (p + r)))\ *100 \tag{4}$$

The $a$ percentile represents the total number of correct detections. The $p$ percentile indicates the ratio of positive predictions performed correctly based on all positive predictions (including false ones). The $r$ percentile is used to access the ratio of correct positive-predictions in relation to all positive predictions. The F1 Score ($F$) is the harmonic mean between $p$ and $r$, i.e., the mean between the errors of commission and omission; higher $F$ values indicate higher agreement between gaps identified in the imagery data (observed value) and that were validated in the field (true value).

#### 2.5.2 Gap geometry

We quantified gap height and area from the three-dimensional structure of the forest canopy. Gaps formed during the period for which only the UAV monitoring was available, were validated during a single field-campaign. The area of these gaps was also measured according to Brokaw's (1982) method. We tested how height loss was correlated with the area of the gaps using Pearson's correlation. We used paired t-test to compare gap geometry calculated from our UAV imagery and field data.

We used both UAV imagery and field data to describe the size distribution of gaps. We then fitted three probability distributions: Exponential, Power-law (or Pareto), and Weibull to determine which best described the size distribution of observed gaps. The parsimony of the fitted models was assessed using the Akaike information criterion (AIC) (Burnham and

Anderson, 2002). We also assessed the best fit using the Kolmogorov-Smirnov statistic to compare the maximum difference in the cumulative probability distributions between the observed and the fitted data (Carvalho, 2015). Fits were obtained by using absolute values of frequency (Araujo et al., 2021a). We tested the size class distributions from the smallest gap size found in both methods (9 m$^2$ and 10 m$^2$, for field data and UAV imagery, respectively). We also fitted the distribution model only for gaps $\geq 25$ m$^2$ to test for possible differences related to the relatively higher proportion of small-sized gaps in our data set.

### 2.5.3 Mechanisms of gap formation, biomass losses and structure of gaps

Combining high-resolution remote sensing with forest inventory data allowed us to identify and differentiate between gaps formed by the death of single trees, tree clusters and branch fall. We counted and determined the area of gaps formed by each of these mechanisms. The main mode of tree mortality was determined from detailed observations as described in subsection 2.3.1. We tested for possible differences in area and released biomass among mechanisms of gap formation using Analysis of Variance (ANOVA); p-values were computed based on two-tailed.

### 2.5.4 Correlations between gap frequency and area with precipitation

We assessed the correlation of gap frequency and area with cumulative precipitation and extreme rainfall events using the imagery data. Initially, we calculated the rate of gap area formation by dividing the summed area of all gaps by the duration (in days) of respective time intervals during which the gaps were observed (11-80 days). We further adjusted these to express the rate of gap area formation in hectares per month. Gap frequency rate was also computed from the summed area over the different time intervals, and was expressed in hectares per month. The temporal variation of gap area and frequency were normalized by the time in months between each pair of images. We correlated these variables using Pearson correlation.

### 3 Results

### 3.1 Sensitivity of gap detection

We detected 32 gaps formed between September 2018 and January 2021 (Fig. 3). Out of that, 14 gaps were formed during the monitoring period for which no simultaneous field data were acquired. Another 18 gaps were formed during the period for which we conducted both remote and field monitoring (November 2019 to January 2021).




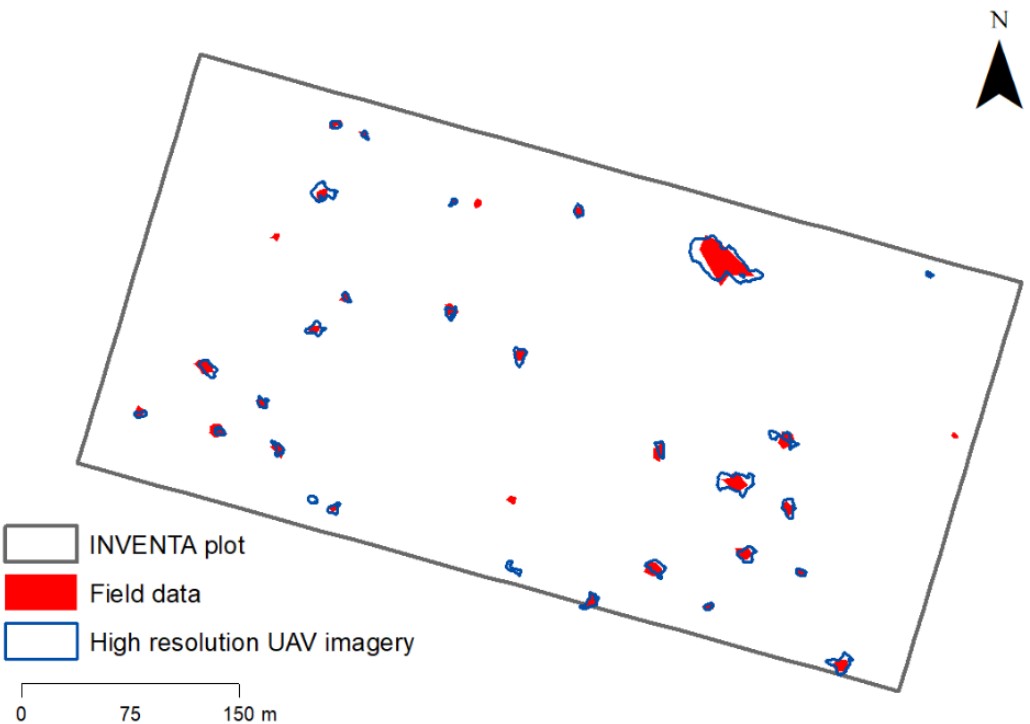

**Figure 3: Map including the location of canopy gaps identified with UAV photogrammetry and inventory plot surveys ("field data") in the INVENTA plot (total area of 18-ha) located in Central Amazon, Brazil, during the period from 18th September 2018 to 19th**
**January 2021.**

For the 18 gaps for which field (true value) and UAV data were available, 14 gaps were detected using both methods; three gaps were only detected in the field and one was only detected in the imagery (Table S2). The accuracy, precision, recall sensitivity and F1 score obtained with our remote sensing UAV method were 77.78 %, 93.33 %, 82.36 % and 87.50 %,
respectively.

The three gaps detected exclusively from field data were formed by the fall of standing dead trees (total area of 15 m$^2$ and 26 m$^2$) and branches (20 m$^2$). These gaps were not visible on either the difference images or the orthomosaics, which indicates that there was no traceable change in the upper canopy of the forest. The single gap only detected from imagery data was formed by the partial loss of the crown of a standing dead tree. Importantly, this gap does not fit the definition by Brokaw
(1982), in which gaps are considered as openings that extends from the upper canopy to the understory (i.e., at least two meters above the ground).



## 3.2 Patterns of gap geometry

The size of gaps identified from imagery and field data varied from 10.37 $m^2$ to 834.65 $m^2$ and from 9.59 $m^2$ to 580.65 $m^2$, respectively (Table 1, Fig. S2a). The differences between the smallest and largest gaps detected with the two methods were

1.39 $m^2$ and 254 $m^2$, respectively. Our data provide no evidence for strong differences in gap area between methods (p= 0.8544). Nonetheless, gap perimeter and shape complexity index (GSCI) varied significantly between methods (p= 0.01019 and p ≤ 0.001, respectively) (Table S3).

Approximately 50 % of gaps described within the 28 months for which field and imagery data were available had total area ≤ 40 $m^2$. This result indicates that in our study site, relatively small gaps are the most frequent canopy disturbance (Fig. 4, and

Table S4). Although more frequent, these relatively small disturbances accounted for only ~16 % of the cumulated gap area. Gap size was positively related to reductions in canopy height (Pearson r= 0.64; p= 0.0003) (Fig. 5). The two most discrepant gaps (area of 36.76 $m^2$ and 14.02 $m^2$ and mean height loss of 1.13 m and 2.13 m, respectively) were only detected in the field and without prior systematic classification.

**Table 1.** Geometric attributes of gaps detected over a period of 28 months in the INVENTA plot, Central Amazon, Brazil.

| Method | Number of gaps | Size range ($m^2$) | Mean gap size ($m^2$) ± IC (95 %) | Median gap size ($m^2$) | Mean gap perimeter (m) ± IC (95%) | GSCI[1] Mean/Max | Gap fraction[2] (%) | Annualized gap fraction (% year$^{-1}$)[3] |
|---|---|---|---|---|---|---|---|---|
| Field Data | 31 | 9.59 - 580.65 | 68.50 ± 37.91 | 44.88 | 29.86 ± 6.92 | 1.13/1.35 | 1.09 | 0.60 |
| UAV Imagery | 30 | 10.37 - 834.65 | 80.07 ± 56.81 | 37.43 | 35.42 ± 9.22 | 1.28/1.6 | 1.36 | |

1- Gap Shape Complexity Index (GSCI = perimeter / sqrt (area 4 π)), whose smallest reference value is 1.0 for describing a circle (Getzin et al., 2012, 2014); 2- Gap fraction is given by the sum of the area of gaps identified over the studied period of 28 months divided by the total monitored area (i.e., INVENTA plot / 18 ha); 3- Annual gap fraction is given by the sum of the area of identified gaps in an annual basis, (i.e., INVENTA plot / 18 ha / duration of study).






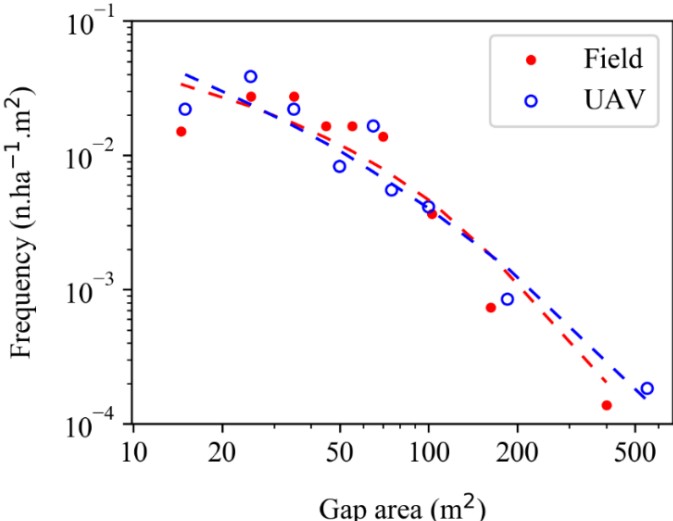

**Figure 4: Size distribution of gaps formed in the INVENTA plot, Central Amazon, Brazil, over the period from 18th September 2018 to 19th January 2021. Gaps were measured from inventory plot surveys (red) and UAV imagery data (blue). Both data sets were fit using a Weibull function (dotted lines).**

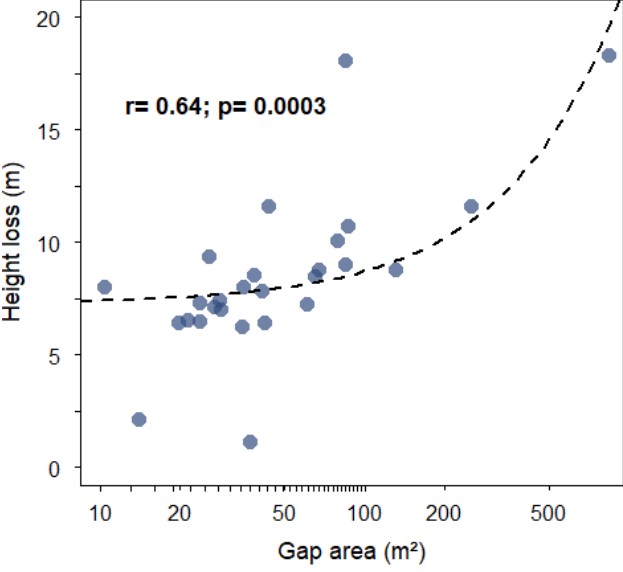


**Figure 5: Relationship between mean canopy height loss and gap area for gaps formed in the INVENTA plot, Central Amazon, Brazil, over the period from 18th September 2018 to 19th January 2021. Gap area was calculated from the UAV Imagery data. The x-axis is log-scaled.**





For both imagery and field data, the distribution of gaps larger than 25 m² was better described by Weibull and Power-law functions (Table 2, Fig. 4). The Weibull function also captured more precisely the size distribution of gaps larger than 9 m² (field data) and 10 m² (UAV data) (Table 2, Fig. 4, Fig S2b, Fig S2d).

**Table 2.** Summary of fitting measures of the exponential, Power-law, and Weibull functions describing the size distribution
of gaps identified on the INVENTA plot, Central Amazon, Brazil.

| Detection method | Minimum size (m²) | Distribution | λ (95 % CI) | α (95 % CI) | K-S | Log likelihood | ΔAIC |
|---|---|---|---|---|---|---|---|
| UAV imagery | 10 | Exponential | 0.014 (0.008 - 0.031) | | 0.2489 | -152.989 | 7.8113 |
| | 10 | Power-law | 1.650 (1.529 - 1.831) | | 0.2242 | -152.035 | 5.9038 |
| | **10** | **Weibull** | **0.512 (0.266 - 1.460)** | **21.544 (0.770 - 65.311)** | **0.1167** | **-148.08** | **0** |
| | 25 | Exponential | 0.013 (0.007 - 0.032) | | 0.2879 | -117.622 | 13.41204 |
| | **25** | **Power** | **2.094 (1.901 - 2.501)** | | **0.07605** | **-110.92** | **0** |
| | **25** | **Weibull** | **0.157 (0.078 - 1.685)** | **0.0002 (0.0000 - 62.147)** | **0.086852** | **-110.747** | **1.661607** |
| Field data | **9** | **Exponential** | **0.017 (0.009 - 0.032)** | | **0.1658** | **-146.783** | **0.3429** |
| | 9 | Power-law | 1.614 (1.494 - 1.766) | | 0.2659 | -153.006 | 12.7899 |
| | **9** | **Weibull** | **0.744 (0.487 - 2.236)** | **40.370 (17.893 - 68.309)** | **0.1462** | **-145.61** | **0** |
| | 25 | Exponential | 0.017 (0.008 - 0.0418) | | 0.214701 | -111.135 | 3.714302 |
| | **25** | **Power-law** | **2.137 (1.797 - 2.710)** | | **0.157811** | **-109.306** | **0.056485** |
| | **25** | **Weibull** | **0.414 (0.131 - 3.004)** | **4.703 (0.000 - 68.937)** | **0.14364** | **-108.28** | **0** |





ΔAIC- AIC (Akaike information criterion) differences to the best model; The best fit models for each data-set are highlighted in bold.

### 3.3 Mechanisms of gap formation and structure, and released biomass

Branch fall was the most frequent mechanism of gap formation, accounting for 34.38 % (n= 11) of all detected gaps (Table 3).
However, the total area accumulated by these gaps accounted for only 17.01 % of the total disturbed area. While gaps formed by tree snapping had the second highest frequency (n= 8 or 25 % of the total number of detected gaps) (Table 3), this mechanism accounted for 59.1 % of the total disturbed area. This result indicates that tree snapping was the most important mechanism of gap formation in respect to the overall disturbed area (Table 3). Uprooting and the fall of standing dead trees were the third and fourth most frequent mechanism of gap formation accounting for 16.53 % (n= 7) and 7.37 % (n= 6) of the
total disturbed area, respectively (Table 3).

Branch fall, uprooting, snapping and standing dead trees accounted for the 52.9 %, 10 %, 6.7 % and 10 % of number of gaps detected on the imagery, respectively. For gaps only identified from field data, these mechanisms accounted for 60 %, 10.3 %, 6.9 % and 3.4 %, respectively.

We found no clear differences in the area attributed to gaps formed by branch fall and the described tree-mortality modes (p=
0.179) (Fig. 6a). However, we found strong evidence that the biomass released in gaps formed by tree snapping was higher than that associated with gaps formed by branch fall (p= 0.0133) (Fig. 6b).

**Table 3.** Relative contribution of the different mechanisms of gap formation observed in the INVENTA plot, Central Amazon,
Brazil, between September 2018 and January 2021.

|  | Gaps (number) | Proportion of gaps (%) | Total area (m²) | Proportion of total area (% m²) |
| --- | --- | --- | --- | --- |
| **Branch fall** | 11 | 34.38 | 414.57 | 17.01 |
| **Snapped dead** | 8 | 25.00 | 1440.58 | 59.10 |
| **Uprooted dead** | 7 | 21.88 | 402.90 | 16.53 |
| **Standing dead** | 6 | 18.75 | 179.55 | 7.37 |
|  | **32** |  | **2437.60** |  |





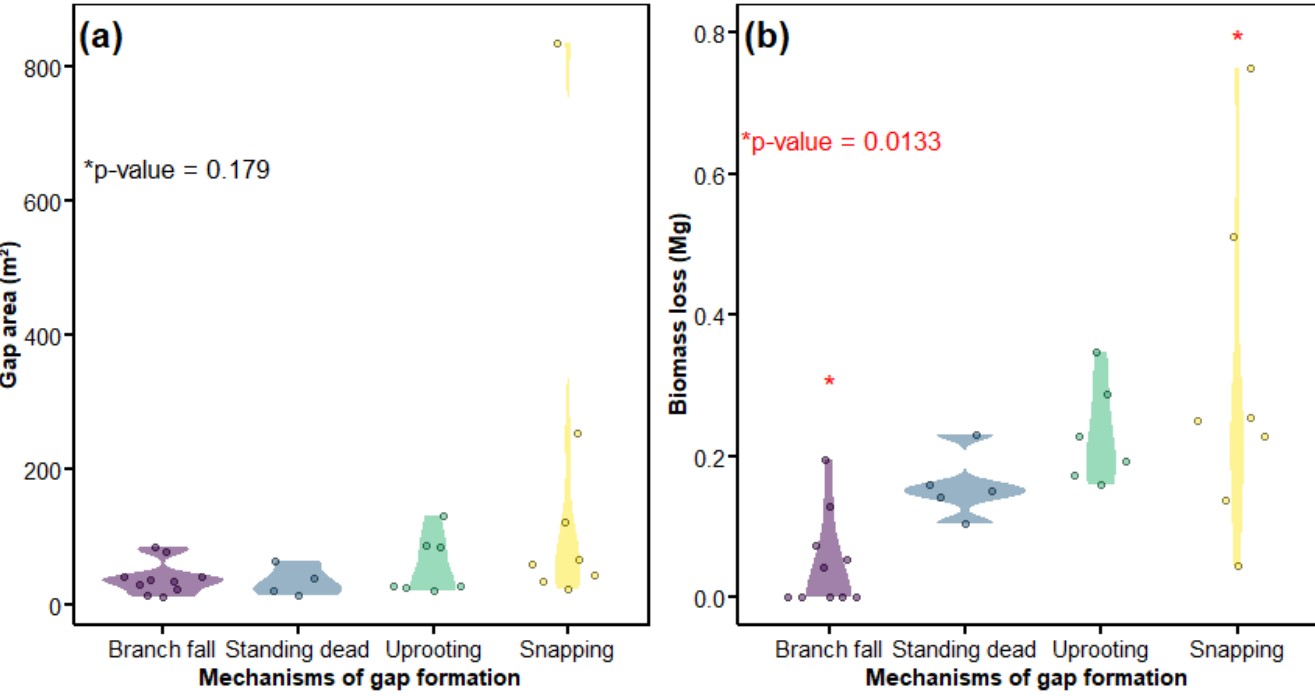

Figure 6: Gap area (a) and biomass loss (b) for mechanisms of gap formation studied in the INVENTA plot, Central Amazon, Brazil, over the period from 18th September to 19th January 2021. The area of gaps was calculated from the UAV Imagery data. We detected significant differences in biomass loss only for branch fall and snapping (panel b).

## 3.4 Rainfall seasonality and gap formation

The gap frequency and area rates were calculated using all 32 gaps identified during the studied period. Although gap frequency and area rate varied among the 28-month-period of monitoring, our data do not support that monthly rainfall influenced these metrics (p= 0.8081 and p= 0.4193; Fig. 7a and b, respectively). However, our data show that monthly gap-formation was marginally correlated with gap area rate for days with extreme rainfall events (i.e., above the 99th percentile, 67.08 mm day$^{-1}$) (r= 0.37 and p= 0.058) (Fig. 8). The time interval accumulating the largest gap area (October 24, 2018 to December 27, 2018) included two extreme rainfall events: 104 mm day$^{-1}$ on 20th October, and 76 mm day$^{-1}$ on 8th November 2018 (Fig. 8).



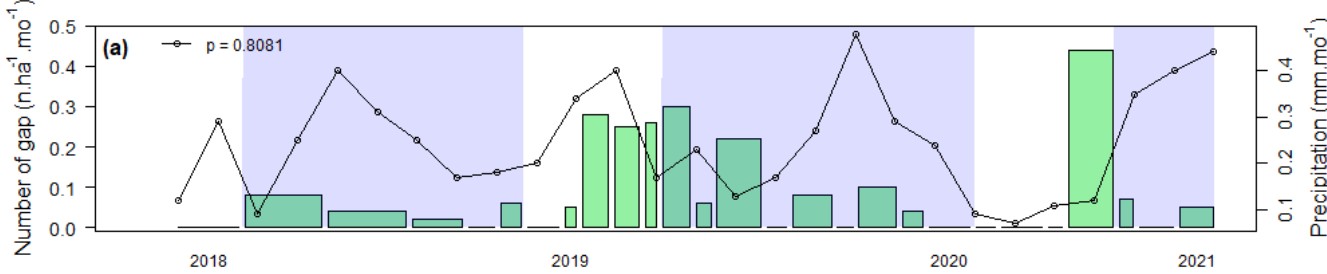

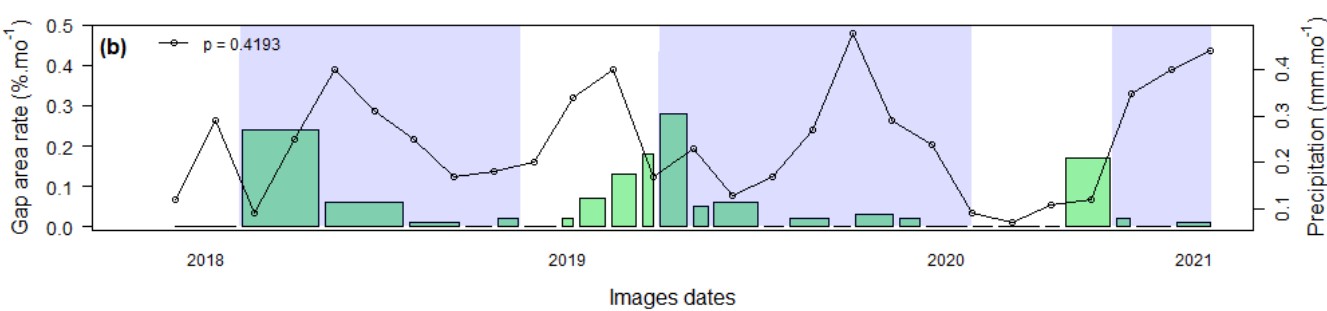

**Figure 7: Seasonality of canopy gaps formed in INVENTA plot, Central Amazon, Brazil, during the period from 18th September to 19th January 2021. Gap frequency (a) and the cumulative rate of gap area formed over the observation period (expressed as % of the 18-ha study area per month) (b). The y2 axis (right) is the cumulative precipitation for each pair of time intervals between images (straight line with dots). The blue shading indicates the rainy season (September to June) for each year. The total area of each green rectangle is proportional to the total area of the gaps formed during the respective interval.**

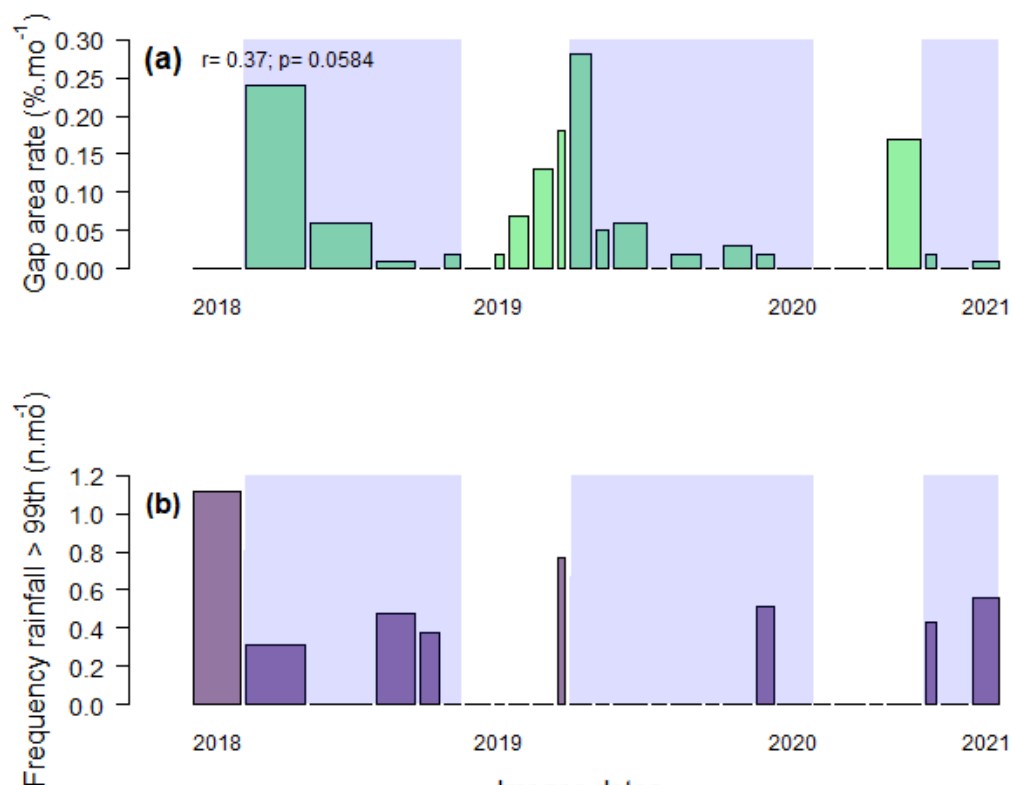

**Figure 8: Cumulative area of gap formation per observation interval (gap area rate per month), expressed as the percentage of the**
**total study area area normalized to a 30-days interval; (a) and frequency of extreme rainfall events per month (b) at the INVENTA**
**plot, Central Amazon, Brazil, during the period from 18th September 2018 to 19th January 2021. The blue shading indicates the**
**rainy seasons (September to June) each year. The frequency of extreme rain events is given as absolute values normalized to a**
**monthly rate for a 30-days interval. The area of the green rectangle (a) is the proportional to the total area of gaps formed between**
**respective observation dates.**

**4 Discussion**

**4.1 Imagery and field data have different sensitivity for detecting small gaps**

We detected 17 and 16 gaps from field and imagery data, respectively; 14 gaps were identified from both methods. However,
it is still possible that our approach underestimated the frequency of canopy disturbances smaller than the size threshold we
analyzed (i.e., 5 m²). In a few cases, gaps detected from UAV imagery data (i.e., losses of canopy height) could not be detected





from field surveys and did not fit the classical definition proposed by Brokaw (1982). If exclusively based on this definition, our results would have overestimated gap frequency by 11.7 %.

Gaps observed in the field but not captured on the imagery were relatively small and mainly formed by the fall of branches from live and standing dead trees. It is important to note that the mechanism of gap formation is related to the sensitivity of detection (Putz et al., 1983; Chao et al., 2009). In particular, branch fall of live and standing dead trees impacted relatively

smaller areas.

Detecting small gaps mainly affecting the forest understory is challenging. Nonetheless, continuous forest inventory surveys allow for the detection of these gaps. Thus, even if no clear signs of opening in the upper canopy are visible from imagery data, minor disturbances can be accounted for with detailed forest inventory. Although understory gaps have lower associated losses of biomass, they can promote changes in light quality and availability, which lead to effective and persistent effects on

the natural regeneration, structure and species composition of tropical forests (dos Santos et al., 2020; Romell et al., 2009; Dupuy and Chazdon, 2008; Magnabosco Marra et al., 2014b).

To our knowledge, this is the first study quantifying biomass losses associated with understory gaps. Here we demonstrated that these gaps contribute relatively little to landscape patterns of biomass. Still, future studies are required to address their importance to processes regulating patterns of species distribution and diversity.

**4.2 UAV photogrammetry is a robust method for monitoring gap dynamics in Amazon forests**

Our tests comparing gaps detected from UAV photogrammetry and field data are rarely found in the literature (Yue et al., 2019), especially for dense tropical forest. Although understory gaps can be missed, the results of our research confirm the suitability and robustness of UAV photogrammetry for monitoring canopy dynamics in closed-canopy forests. When combined with continuous forest inventory, UAV photogrammetry at high temporal and spatial resolution can also reveal associated

mechanisms of gap formation and released biomass.

However, the differences we found in perimeter and GSCI between imagery and field data indicate that geometric attributes of gaps identified remotely and in the field can diverge. In our study, the differences between these methods are likely due to describing field-identified gaps as polygons that always had eight vertices. This contrasts with our remote estimates, on which losses of height (z value) and gap geometry were computed from 1 m$^2$ pixels and for polygons which had a varying number of

vertices. The shape of gaps measured in the field tended to be elliptical (Runkle, 1981) and triangular (Eysenrode et al., 1998). To date, most methods describing the shape and area of gaps focused on a two-dimensional projection of the canopy to the forest floor. In these two-dimensional assessments, there are three main assumptions: (i) most of the gaps have an uniform elliptical shape; (ii) the shape of irregular gaps can be approximated with several measurements; and (iii) the area of irregular-shaped gaps can be only be calculated from hemispherical photos (Schliemann and Bockheim, 2011). Here, we applied high-

resolution imagery to assess gap geometry more detailed and beyond the number of vertices commonly applied in traditional field measurements. Vepakomma et al., (2008) combined LiDAR point cloud and field data from a boreal forest and also





reported great differences in the shape of gaps derived from these two approaches. According to these authors, the more complex the shape and perimeter, the greater is the difference between the remote and field measurements (i.e., ground truth). Although the area of the gaps did not vary between our two methods, the reported variations in perimeter and GSCI revealed that imagery data allow for more complex shapes that can better represent natural disturbances (Lertzman and Krebs, 1991; Gagnon et al., 2004). This was true for our study region, for which gaps detected from imagery data had a greater variety of shapes, often irregular. The shape is an important feature for understanding the structure and dynamics of tropical forests (Jucker, 2022), which is important for determining microsite resource availability (Canham et al., 1994) from the center to edge of gaps (Gagnon et al., 2004). For the Amazon, there is still little research on how the shape of gaps varies across environmental and disturbance gradients (Malhi and Román-Cuesta, 2008).

The higher frequency of relatively small gaps we report here corroborates other studies that used different detection and classification methods (Lawton and Putz, 1988; Brokaw, 1982; Yavitt et al., 1995; Vepakomma et al., 2008; Asner et al., 2013; Leitold et al., 2018; Dalagnol et al., 2021). A Power-law exponent ($\lambda$) < 2 reflects a large proportion of smaller gaps (Asner et al., 2013). Our results indicate that remotely detected gaps larger than 25 m² follow a Power-law distribution ($\lambda$ = 2.137 ± 0.913, mean ± CI 95 %, respectively). This agrees with studies that focused on the size distribution of relatively larger gaps detected from coarser-scale remote sensing data (Fisher et al., 2008; Chambers et al., 2009). Regardless of differences in forest structure, climate, and disturbance history among the regions on which these studies were conducted, $\lambda$ appears to converge on a narrow set of values for different tropical forests (Jucker, 2022). Using LiDAR data from 421 sites in the Brazilian Amazon, Reis et al. (2021) showed that $\lambda$ varies between 1.66 and 2.50 across the basin, mainly reflecting gradients in underlying tree mortality, canopy height, and human disturbances. However, prior to our study, no model supported with field data had yet been fitted for Amazon.

In our study site, however, the frequency of gaps larger than 10 m² was better captured by a Weibull function. This can be explained by the size threshold we use for defining our gaps. The relative lower density of small canopy disturbances compared to what would be expected under a power function may be partially explained by lower detection frequencies, i.e., measurement bias (Araujo et al., 2021). This may be more important for gaps < 10 m². Still, our results show that independent of the detection method, the best fit describing the size frequency of gaps from 9 m² to 835 m² in our study region was achieved with Weibull. As confirmed by our detailed forest inventories over the imaged period, we believe that this pattern was not biased by gaps eventually not detected from imagery data.

## 4.3 Small-scale disturbances dominate canopy dynamics and associated biomass losses in Central Amazon

Repeated and field measurements provide allow for quantifying the relative importance of mechanisms of gap formation in Amazon forests. Our results resemble a previous study using repeated high-density Lidar data on another location in Central Amazon, Santarém, Pará (Leitold et al., 2018). These authors showed that biomass losses due to single and multiple events of branch fall events accounted for only 20% of the total estimated biomass loss from canopy and understory trees. In Panama,



branch fall accounted for 43.5 % of the gap density over a period of five years, but only for 23 % of the total disturbed area
(Araujo et al., 2021). Like in our study region, this pattern highlights that the size and shape of gaps is largely influenced by
modes of tree mortality. In tropical forests, the mortality rate of trees from 1 cm to 10 cm DBH was unrelated to tree biomass
losses among trees > 10 cm DBH (Gora and Esquivel-Muelbert, 2021). Still, there is a relative contribution of different tree
mortality factors across a continuum of tree sizes (Gora and Esquivel-Muelbert, 2021). Zuleta et al. 2022 showed that uprooted
trees have significantly larger size (i.e., DBH). However, snapping was a more frequent mode of tree mortality compared with
standing dead or uprooting. Although of less importance among large trees, falling branches can affect small trees differently
and promote changes or filter out saplings of canopy and also understory species.

Crown damage and/or loss is one of the most impactful risky aspects leading to tree mortality (Zuleta et al., 2022). If climate
change results in a higher frequency of storms and extreme winds, branch fall and thus tree mortality rates can also be expected
to increase. This may affect carbon stocks and dynamics, as well as the functional composition of these forests at the landscape
level (Magnabosco Marra et al., 2018; Denslow et al., 1998).

### 4.4 Extreme rainfall-events control gap formation

In our study site, the gap area and frequency rates varied over time. The gaps formed during a single period of less than a
month (21st October to 1st November 2020) accounted for 20.4 % of the total disturbed area. Still, we did not find a correlation
between gap area and frequency rates with the accumulated precipitation over time. Fontes et al. (2018) reported a strong
positive correlation (r= 0.85) of cumulative precipitation and tree mortality over a 1-year period on a plot contiguous to our
study site, which may be related to interannual variability of the rainfall (Marengo et al., 2009). Nonetheless, we found a
positive correlation between gap area rate and the frequency of extreme rainfall events. As in our study site, the frequency of
rainfall events above the 98th percentile (24.3 mm hour[1]) explained a large fraction of the variation in rates of gap area over
measurement intervals (r = 0.46) for a tropical forest in Panama (Araujo et al., 2021).

As recently reported for the Amazon, areas with stronger winds and more frequent lightning have larger gaps (Reis et al.,
2021). Extreme winds and rain can cause extensive damage (single gaps >10 ha) in the forest (Negrón-Juárez et al., 2018;
Espírito-Santo et al., 2014; Magnabosco Marra et al., 2014a), but the size distribution and landscape effects of small-scale
storm-related disturbances are more challenging to study. Convective rainfall and extreme wind gusts promote crown damage,
snapping and uprooting from individual to large clusters of trees (Magnabosco Marra et al. 2014a; Negrón-Juárez et al. 2011;
Nelson et al. 1994). The vulnerability of trees to extreme wind and rainfall vary across Amazon regions (Negrón-Juárez et al.,
2018; Urquiza Muñoz et al., 2021). Thus, projected shifts on the intensity and frequency of these events can also be expected
to have particular effects on current patterns of tree mortality and biomass.

In addition to seasonal patterns of rainfall and wind, gap formation is also affected by local topography and soil (De Toledo et
al., 2011). In Central Amazon forests, despite little variation associated directly with soil and slope, tree mortality due to
uprooting and snapping can increase with more frequent storms (De Toledo et al., 2012). As climate change is expected to

alter the frequency and intensity of tropical storms, soil attributes and topography may become more useful to improve estimates of tree mortality and biomass losses over large areas in Amazonia.

## 5 Conclusion

By combining high temporal and spatial resolution UAV imagery with detailed field data on the mechanisms of gap formation, we could reliably assess the geometry and relative importance of canopy gaps for a closed-canopy Amazon forest. Furthermore, this integrated approach allowed us to relate geometry and size patterns with different mechanisms of gap formation. Although a larger proportion of canopy gaps could be detected from orthomosaics, their mechanisms of formation could only be distinguished using field data. Thus, our study supports that detailed forest inventories are fundamental for evaluating remote sensing products and metrics.

The Weibull was the most appropriate function for describing the frequency distribution of gaps in our study region. However, the relative importance of mechanisms of gap formation may change as a function of climate change, and consequent shifts in the rainfall and extreme wind gusts. The patterns described here may vary from region to region and along topographic gradients, which was beyond the goals of our study. Although our sampling effort was sufficient to detect and describe major mechanisms of gap formation, further studies are required to generate proxies for distinguishing mechanisms of gap formation

related to extreme weather events such as storms and lightning increasing rates of snapped and uprooted trees, or severe droughts that are more related to standing dead mortality. These can improve current estimates of carbon balance, in tropical forests, as well as reducing the uncertainty of models of vegetation dynamics.

## 6 Acknowledgments

This study is part of the INVENTA and ATTO Projects funded by the German Federal Ministry of Education and Research

(BMBF, contracts 01LB1001A and 01LK1602A), the Brazilian Ministry of Science, Technology and Innovation (MCTI/FINEP, contract 01.11.01248.00) and the Max Planck Society (MPG). Our study site is also supported by the INCT Madeiras da Amazônia. We thank the Forest Management Laboratory of the National Institute for Amazonian Research (LMF/INPA) for logistic support.

## Author contributions

AS and DMM planned and designed the research; AS, RFA and CHSC collected drone data; AS, RFA, CHSC and FRSS processed drone imagery; AS and FRSS collected field data; AS, DMM, RFA and CHSC analyzed the data; AS and DMM wrote the manuscript with the support from all authors.



**Competing interests**

The authors declare that they have no conflict of interest.

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
