# Peer review of "Gap geometry, seasonality and associated losses of biomass - combining UAV imagery and field data from a Central Amazon forest"

_Biogeosciences, 2022_

## Author Comment (AC1)

**Preprint bg-2022-251**

**Response to Reviewer 1**

The article "Gap geometry, seasonality and associated losses of biomass – combining UAV imagery and field data from a Central Amazon forest" studies gap formation on an 18ha field plot in the Amazon, using both remote sensing (photogrammetry/Structure from Motion) and field data. It provides an interesting look into canopy dynamics at one particular tropical forest site and a comparison (or validation) between field-based methods and remote sensing, which is crucial in linking traditional approaches with modern technology. Due to its substantial field sampling effort, the study can relate gap formation to different tree mortality modes and to associated biomass losses, thus linking ecological processes to the carbon cycle, which should be of great interest to readers of Biogeosciences. I also found the paper generally very well written, with well thought-through methods and clear and concise descriptions.

There are, however, a few changes/issues that I would recommend the authors to consider before publication. I will highlight a few larger aspects first, and then provide line-by-line comments in a classic review style.

*We thank Reviewer 1 for this supportive revision and constructive comments. We will consider all aspects for preparing our revised manuscript. Please, find below a point-by-point answer to all questions.*

**1/ Definition of gap**: My impression is that the definition of gaps is not entirely consistent in the study. On the one hand, Brokaw's definition of gaps as extending down to 2m in canopy height seems to be used (l.161), but on the other hand, the authors argue several times that there are undetected "understory gaps", or gaps that are not visible in the upper canopy. Specifically, they attribute the differences between UAV imagery and field data to the UAV imagery not being able to detect such subtle changes below the canopy. But if we use the Brokaw definition, that should not be the case, as any gap would, by necessity, be a hole in the upper canopy and extend down to the ground, no? Could it be that the authors implicitly use treefall events or other canopy characteristics as part of their gap definition in their field-based studies? Could this also explain why gaps created by standing dead trees were the main difference? The definition aspect also affects what should be considered the "truth" for the validation – field-based assessments certainly offer more information to interpret gap formation (is it a branch fall or a tree fall? etc.), but to automatically consider them the truth (l.209) is not evident to me. Could one not argue that the 3D canopy height models derived from photogrammetry (or even better, lidar) can more accurately quantify height changes than visual/manual assessments?

*As suggest by Reviewer 1, we will clarify the definitions of gap and revise the text for consistence.*

*We used the definition by Brokaw (i.e., gap in the forest canopy extending from the upper stratum to an average height of two meters above ground) to compute and measure gaps in the field* (Brokaw, 1982). *This is a classical and efficient method, which allows comparing our findings with those from fundamental work conducted in other tropical forests.*

*Gaps were identified through detailed inspection of dead trees and fallen branches. For the identification of mortality events and description of mortality modes, we followed the protocol of previous studies developed in our study region (Marra et al. 2014, 2018; Negron-Juarez et al. 2011; Ribeiro et al. 2016). To minimize errors, the inspections were conducted across relatively narrow strips of forest (less than 10 m), and were always led by ASLP. A field campaign was undertaken to quantify and mark all pre-existing gaps (i.e., baseline).*

*Our remote-sensing approach provides detailed data on the upper canopy of the forest, but no information on the understory. Gaps were defined as disturbed patches with total area $>5\ m^2$ and with reductions of canopy height greater than 10 m. These thresholds were established based on the nominal resolution of our processed imagery (1 m) and the scale at which the forest inventories were conducted, i.e., tree level. Overall, the fall of branches and/or standing dead trees produced severe damage mostly in the upper canopy while the understory remained intact. Therefore, upper canopy gaps detected remotely were not always detected on the ground using the definition by Brokaw. This pattern shows that apart from covering relatively large areas at low costs, UAV photogrammetry is an efficient method for monitoring gap dynamics - including the detection of upper canopy disturbance usually not visible from the forest ground.*

*We believe our approach combining field with remote sensing data provides interesting insights on concepts and methods for quantifying gaps and their effects on forest dynamics. Classical methods based on field observations are efficient to detect gaps extending from the upper canopy to the understory of the forest, and are crucial for validating remote tools, and for quantifying and modeling associated losses of biomass. High-resolution photogrammetry allows for more precise measurements of the features of the gaps, including those restricted to upper canopy and/or causing minor damage. Although more frequent, small-scale disturbances not implying tree mortality such as defoliation and branch fall are often neglected in forest inventories. Our study brings novel knowledge on the contribution of these events to ecosystem processes such as carbon cycle. Quantifying the size-distribution of gaps and their landscape importance is crucial to understand how forests respond to shifts in the disturbance regimes trigged by climate change and land use.*

**2/ Study area size and gap size frequency distributions (GSFD):** Having such a detailed comparison between field based and remotely sensed gap structure is an important feat, based on substantial field work, so it makes sense that the authors focused on a plot size of "only" 18ha. However, this limits the analysis somewhat when it comes to assessing GSFD and the "landscape scale" patterns the authors are interested in. As expected for 18ha, sample sizes are very small (32 gaps in total, but only 14 gaps that co-occur in both field and remote sensing data). I am sceptical that such sample sizes yield much information on which distribution actually fits better, and I would expect the fitted Weibull, exponential and power law distributions to be so uncertain in their parameters (the power law exponent has an uncertainty of 2.137 +- 0.913, which is huge) that there is not much sense in comparing the fit of different GSFDs (one single data point might already shift the goodness of fit). If the authors would like to keep this analysis, I suggest they explicitly use confidence intervals / simulations of data generation to assess how reliably these distributions can actually be differentiated with so few gaps, or maybe focus less on which distribution fits better and more on the field-remote sensing comparison. They should also provide a careful discussion that does not place too much emphasis on the different AIC values (which have generally low delta, anyways). More generally, if this type of analysis is carried out, I would also highly recommend the additional fitting of a lognormal distribution, which comes about through similar generative processes as power law models and is often an equally good fit.

*We thank the Reviewer 1 for this valuable comment. We used a bootstrap with 1000 interactions for calculating the confidence interval of the different fits and will include the information in the revised figure. We also tested lognormal as part of our analyses.*

*bsfitsizedistcontdata <-*
*function(sizedf,minsize=c(10,25),maxsize=c(720,900,NA),fitfcn=c("exp","pow","weib","logn"),*
*nbootstraps=1000,alpha1=0.05, alpha2=0.01)*

*#   nbootstraps is the number of bootstrap interactions to run to get CIs*

*We also used the Kolmogorov-Smirnov statistic to evaluate the goodness of fit (not only the AIC). We also tested lognormal as part of our analyses. The lognormal distribution did not fit for values greater than 25 m$^2$ in the UAV data, and the K-S statistic was also adequate (these are the largest values for a good fit). As requested by the reviewer, we propose to replace the following Figure S2 and Table S2:*

[Figure]

**Figure S2.** Size distribution of gaps at the INVENTA plot, Central Amazon, Brazil, for the period from September 18[th], 2018 to January 19[th], 2021. Size distribution of gaps detected with UAV photogrammetry (a) and field surveys (c). Modeled distribution of gaps larger than 10 m$^2$ detected with UAV imagery (b) and field data (d).

**Table 1.** Summary of fitting measures of the exponential, Power-law, Weibull and Lognormal functions describing the size distribution of gaps identified on the INVENTA plot, Central Amazon, Brazil.

| Detection method | Minimum size (m²) | Distribution | λ (95 % CI) | α (95 % CI) | K-S | Log likelihood | ΔAIC |
|---|---|---|---|---|---|---|---|

| | | | | | | |
|---|---|---|---|---|---|---|
| UAV imagery | 10 | Exponential | 0.014 (0.008 - 0.031) | | 0.2489 | -152.989 | 9.888 |
| | 10 | Power-law | 1.650 (1.529 - 1.831) | | 0.2242 | -152.035 | 7.981 |
| | 10 | Weibull | 0.512 (0.266 - 1.460) | 21.544 (0.770 - 65.311) | 0.1166 | -148.08 | 2.077 |
| | **10** | **Lognormal** | **3.551 (2.137 – 3.995)** | | **0.8466** | **-147.04** | **0** |
| | 25 | Exponential | 0.013 (0.007 - 0.032) | | 0.2879 | -117.622 | 13.41204 |
| | **25** | **Power** | **2.094 (1.901 - 2.501)** | | **0.0760** | **-110.92** | **0** |
| | 25 | Weibull | 0.157 (0.078 - 1.685) | 0.0002 (0.0000 - 62.147) | 0.0868 | -110.747 | 1.661607 |
| | 25 | Lognormal | - | - | - | - | - |
| Field data | 9 | Exponential | 0.017 (0.009 – 0.032) | | 0.1658 | -146.783 | 4.9063 |
| | 9 | Power-law | 1.614 (1.494 – 1.766) | | 0.2659 | -153.006 | 17.3533 |
| | 9 | Weibull | 0.744 (0.487 – 2.236) | 40.370 (17.893 – 68.309) | 0.1463 | -145.61 | 4.5635 |
| | **9** | **Lognormal** | **3.742 (3.3510 – 4.0823)** | | **0.8368** | **-143.33** | **0** |
| | 25 | Exponential | 0.017 (0.008 – 0.0418) | | 0.2147 | -111.135 | 4.5447 |
| | 25 | Power-law | 2.137 (1.797 – 2.710) | | 0.1578 | -109.306 | 0.8869 |
| | 25 | Weibull | 0.414 (0.131 – 2.982) | 4.703 (0.000 – 68.937) | 0.1437 | -108.28 | 0.8305 |
| | 25 | Lognormal | 3.3953 (-17.2998 – 4.1895) | | 0.9996 | -107.86 | 0 |

*To our knowledge, this is the first study merging field and remote sensing data for assessing the geometry of gaps and computing their contribution to processes regulating carbon cycle in Amazon. Our study region is covered with old-growth terra-firme forests including a topographic/edaphic gradient comprised of plateaus, slopes and valleys. Our 18-ha plot is monitored since the year 2000; all the trees >10 cm DBH are recorded and tagged (~10.500 individuals). Apart from relying on this long-term inventory, our study also used LiDAR and rainfall data available for the same plot. Flying a contiguous area allowed us to reduce costs and optimized the logistics of field campaigns, which was crucial for the success of a 28-months monitoring.*

*Our relatively large plot allowed for a robust quantification of tree mortality and associated losses of biomass across an environmental gradient. A total of 32 gaps were formed during the entire studied period; 18 were formed during a 14-month period for which field and remote data were acquired. However, our results shall not be extrapolated beyond our study region. A regional assessment of the size distribution and geometry patterns of gaps require the inclusion of further sites reflecting existing variations in forest attributes and disturbance regimes. We will improve the discussion on this aspect in the revised version of the manuscript.*

**3/ Precipitation and gap formation:** This part of the paper, while relevant, is not really motivated in the introduction, and more effort should be spent on explaining why it is relevant to suppose a link between precipitation and gap formation, and why presumably more direct drivers of gap formation (wind or even lightning) were not used. It is understandable that such data may not be available, but nothing in the introduction/methods section explains why precipitation is interesting. I would also remove the analysis of extreme rainfall events, because this seems like a filtering of the data that could be done with many thresholds (90th / 95th percentile, etc.), and with only 3 years and 8 data points for extreme rainfall (Figure 8), I doubt that the correlation the authors found tells us much about the system.

*We agree with the Reviewer 1 and will improve the text explaining the links between precipitation and gap formation. In our study region, extreme wind gusts and precipitation are important mechanisms of tree damage and mortality (Chambers et al. 2013, Magnabosco Marra et al. 2018, Negrón-Juárez et al. 2018, 2023). A study monitoring tree mortality over five decades in a Central Amazon forest found that trees died more frequently in wet months, even during drought years (Aleixo et al., 2019). A positive correlation between precipitation and tree mortality was also reported for our study area (Fontes et al., 2018). A regional study based on 12 years of satellite data found that major windthrows (visible on Landsat) in Central Amazon occurred more frequently between September and February, months characterized by heavy rainfall, than the rest of the year (Negrón-Juárez et al., 2017). These background studies support that a greater number of gaps can be expected during the rainy season. We will revise the introduction to clarify this aspect.*

*We also understand that a comprehensive assessment of the influence of precipitation on gap formation requires longer-term data addressing seasonal and interannual variability. While this aspect could not be fully investigated with our 28-month dataset, it is part of an ongoing study which will incorporate a larger monitoring period and thus a higher number of extreme rainfall events and potential associated gaps. We will also aggregate background information on this aspect in the introduction and point out the limitations of our assessment. In addition, we suggest moving Figure 8 to the Supplementary Material.*

**4/ Remote sensing vs. field data in assessing mechanisms of gap formation and biomass loss:** My impression was that section 3.3 would be one of the most interesting sections for readers of Biogeosciences, and that the authors could extend their analysis here a little bit without too much effort. For example, I would relate released biomass to overall plot biomass. There could also be an interesting comparison of released biomass visible from gaps, to overall biomass released from tree mortality, also counting understory mortality (if these data exist). Finally, since they have such a comprehensive data set, the authors could also compare other aspects of gaps between the different mortality modes (branch fall, snapped, etc.). I would suggest a look at the metrics the authors already calculated (gap geometry), but also previous and surrounding canopy height, and maybe also gap closure rates, with a focus on the values from remote sensing. Maybe, the authors could also use the RGB signature of the orthophotos as an additional metric to compare between

mortality modes. Such an analysis would provide some hints on whether remote sensing/photogrammetry could distinguish different modes of mortality/gap formation/biomass losses, or at least separate one specific mode (standing dead). These are only suggestions and would, of course, only be indicative due to the small

sample sizes, but I think they might be very interesting for future studies/Biogeosciences readers and be in line with the authors' objectives to assess how much we can learn from remote sensing compared to field-based assessments.

*Biomass estimations are indeed an important aspect of our study, and we thank the Reviewer 1 for stimulating this discussion. As suggested, we did calculations to compare the biomass released in gaps vs. the stocks of old-growth forests in our study region. Apart from discussing these numbers, we will include a new figure in the Supplement (Fig. S3) to show the size distribution of gaps formed by branch fall and the different modes of tree mortality.*

**Figure S3.** *Size distribution of gaps formed by branch fall and different modes of tree mortality.*

*We agree that our RGB chronosequence can support research beyond the scope of this manuscript. For instance, we are investigating the geometric and reflectance signature of gaps created by branch fall and modes of tree mortality (i.e., standing dead, snapping and uprooting). This investigation may contribute to more reliable assessments of tree mortality events related to opposing climate extremes such as storms and droughts.*

*Our sample unit in this study is a gap event, which is not dependent on the size of the plot or subplot size. The18-ha plot was selected because (i) it is covered with an old-growth forest; (ii) includes a topographic/edaphic gradient (see answer to the second comment of Reviewer 2); (iii) has an infrastructure of trails and (iv) forest inventory, LiDAR and rainfall data.*

**Line-by-line comment:**

3: Is the title actually accurate? Gap geometry and seasonality do not seem to be such important results/aspects of this study, so maybe rethink/rephrase it?

*We agree and suggest the following title: "Gap attributes and associated losses of biomass - combining UAV imagery and field data from a Central Amazon forest".*

37: What is a multi-temporal process? Maybe rephrase?

*We suggest rephrasing as following: "The maintenance of these stocks depends on dynamic processes that regulate the growth and mortality of trees"*

41: Even though this may not be fully relevant to the paper, maybe droughts could be mentioned as another major extreme event?

*We agree that droughts are another extreme event that can influence gap dynamics by causing water stress and consequently increasing the frequency of branch fall and standing dead trees. We will include a comment on that in the revised version of the manuscript.*

50: This may be a definition question and not crucial, but in the context of tropical forests, gaps that are thousands of hectares in size (or tens of squarekilometers) seem unlikely, or probably not what tropical ecologists would commonly classify as gaps (e.g. one or several large canopy trees falling and leaving a gap in the canopy). Such a definition seems more common in fire-dominated boreal ecosystems. Maybe you could add one sentence specifically on tropical gap sizes. Also, this would be more in line with the extent of your sample plot.

*We thank Reviewer 1 for this important comment regarding the definition of gap size. This scenario of giant gaps is also common in the Amazon. Although with some surviving trees, gaps opened by extreme rain and wind in Amazonia can have a total area greater than 3,000 hectares (Nelson et al., 1994; Espírito-Santo et al., 2014; Negrón-Juárez et al., 2010, 2018, 2023). Recent evidences suggest hotspots for the occurrence of blowdowns (Negrón-Juárez et al., 2023; Urquiza Muñoz et al., 2021) and that the frequency of these events is regulated by atmospheric phenomena that are highly sensitive to climate change, such as the potential energy available for convection (CAPE) (Feng et al. 2023). We will include information on that aspect in our revised manuscript.*

62-74: My impression is that this part of the paper jumps quite a lot between points, i.e. from the advantages of remote sensing, citing lidar remote sensing studies such as Dalagnol et al. 2021, to different definitions of gaps, to the problems of optical remote sensing. My question would be: Is the discussion of Landsat needed here, as UAV operates on a very different scale. A more interesting point might be how UAV photogrammetry differs from ALS/UAV lidar (e.g. no within-canopy structure, no ground model, but likely cheaper, more flexible [although limited by meteorological conditions]).

*Our goal here was to discuss how UAV imagery could expand on previous assessments of windthrow tree-mortality, which up to date is mainly based on Landsat imagery (Nelson et al., 1994; Espírito-Santo et al., 2014; Negrón-Juárez et al., 2010, 2018, 2023). Although allowing for long-term studies at the regional scale, Landsat is only sensible for detecting relatively large gaps (>1,000 $m^2$), which in Central Amazon usually involve the death of more than eight trees (Negrón-Juárez et al., 2011; Chambers et al., 2013).*

90: what are "traceable" modes of tree mortality? Or what would be "untraceable" ones?

*We will rewrite the sentence for clarification. Recurrent field surveys following the routine and protocol established in previous studies (Magnabosco Marra et al., 2014, 2018; Ribeiro et al., 2016; Fontes et al., 2018) allowed us to clearly distinguish branch fall, standing dead, snapped and uprooted trees. For example, a standing dead tree may lose part or all of its crown shortly after dying. Thus, when field surveys are not conducted frequently, it is not possible to distinguish*

*the mechanism of gap formation. A snapped tree is characterized by mechanical disruption of the stem (breakage or cracking). If checked in the field within a short period of time after its occurrence, it is still possible to find leaves attached the crown, thin wood/fibers at the snapping point/height and sap (e.g., resins and latex). Uprooted trees have exposed roots usually still attached to the trunk/crown. Branch fall gaps are formed when branches of living trees are broken and/or fall after dying. We will edit the correspondent text for clarification.*

90: The last question with regard to rainfall is very adhoc and not really set up in the introduction. I would provide justification in the introduction on why precipitation should be relevant for gap formation. Would wind be a more important variable?

*Thunderstorms may propagate extreme wind gusts and rain. For instance, Araujo et al. (2021) found a significant relationship between gap formation and precipitation (extreme rainfall events, i.e. the rainfall rate in mm $h^{-1}$ of the 98.2th). We agree with Reviewer 1 and propose including more information to support why we expect precipitation to be an important aspect regulating gap dynamics and geometry.*

113-136: I am no expert in SfM/photogrammetry, but this seems well-described and a good workflow. I have one question: How did you deal with different meteorological conditions during planned flights (fog/rain)? Did you, for example, postpone scans during rainy days? Could this affect your results? How consistent was the timing of the acquisitions on average? I don't think this would be a major problem, but it would be good to mention this somewhere here.

*We thank the Reviewer 1 for this question. The flights were always carried around 09:00h and on the absence of fog and rain. The flights last approximately 15 minutes, which allowed us to acquire images under similar conditions of light. When possible, the flights were carried at cloudy conditions and diffuse light, which improves the visibility of the canopy while reducing shadow. We will add these details in the Supplement.*

155-172: This seems like a substantial effort and great, important work! Just out of interest: since you seem to have access to EBA project's overlapping lidar data, is there a reason why you did not predelineate initial gap distributions from the lidar derived canopy height models?

*The LiDAR data was acquired in 2016. Although at a higher spatial resolution, these data could not be used as a reliable baseline for our study, which started approximately 2.5 years later. Instead, we carried a detailed field survey to collect the coordinates and mark all existing gaps with a plastic-colored stick.*

197-209: This also makes a lot of sense. However, I would move the information from the last sentence (i.e. field value is considered true value) to the beginning to make it clearer to the readers what is considered the validation. I was wondering, however, whether in this case field data can actually be considered the true data? One could make a point that remote sensing (but maybe less so photogrammetry) actually provides a more accurate quantification of the 3D canopy canopy than visual/field-based assessments can. How would you justify your decision?

*We agree with Reviewer 1 and will carry the suggested change. The Brokaw definition was used to identify and describe gaps in the field. Apart from allowing comparison with previous studies, this method is compatible with our measurements of forest structure and losses of biomass. We also used the field data to validate the occurrence of gaps identified and measured remotely.*

215-223: While it is common to fit these distributions and the approach is methodologically sound, does this make sense here? 18ha is a very small area when

it comes to gap delineation, so even without looking at the results, one would assume that your sample size is going to be so low that the inferred distributions are not telling us a lot (and the results bear this out, with 32 or 14 gaps in total). At the very least, I would expect simulations to construct confidence/credibility intervals that show how much variability there is and how uncertain the differences between the different distribution types are. My guess is that it would be very hard to come to any clear conclusion across 18ha. Also, would it not make sense to also test a lognormal distribution? The lognormal distribution is usually the one closest to the power law and comes about through very similar generative processes, so if you fit distributions.

*See answers related to the comment 2.*

225-229: Very interesting! I find the idea of quantifying released biomass very appealing.

*We appreciate this positive comment and agree that the field data is a highlight of our study.*

232: This process of calculating gap area formation rates sounds very complicated. Could you not just take the number/area of gaps that formed between each image acquisition and then divide the number/area by the time between each image? Assuming that images are taken at roughly the same intervals, that should give you a very sound estimate, no? Or am I missing something?

*Apart from relating number of gaps per unit of area, we aimed at analyzing the correlation between precipitation (measured as accumulated rainfall) and gap frequency. We will improve the text for clarification.*

253: "which indicates that there was no traceable change in the upper canopy of the forest". This is probably more a discussion sentence anyways, but I find this problematic. According to the definition (Brokaw) you use, a gap is an "an opening in the forest canopy extending from the upper stratum to an average height of two meters above ground." So by definition something in the upper canopy has to change – either you don't pick it up in the photogrammetry data (maybe one of the processing algorithms is smoothing the canopy too much), or, alternatively, your field-based assessment wrongly found a change in the upper canopy. This could also be an interesting question about gap definitions: should a standing dead tree already be classified as a gap, because light is reaching down almost without obstruction to 2m? How do you interpret this?

*We moved the text to the discussion. A gap easily identified in the field but not recognizable in the images could be one formed by the fall of lower branches from an emergent/canopy tree or the fall of a relatively small standing dead tree. These contribute relatively small area/volume. The respective changes could then be smoothed and gaps not detected. Since dense tropical forests have several vertical strata, the definition by Brokaw only fits when gaps extend to the upper canopy; this was not the case of relatively small damage promoted by branch fall. A standing dead tree was classified as gap when the light penetrated the understory of the forest leading to detectable changes in height/volume.*

260: I'm not sure the p-value is the best way to assess this here. Looking at Figure 3, one would guess that, at the large end of the gap spectrum, UAV seems to find larger gaps than field-based assessments (a difference of ca. 830 m2 to 580m2 for the largest gap seems substantial and larger than I would have expected). How did you

derive the p-value? Did you log-transform the data beforehand (if you assume power-law/lognormal scaling, for example, that would be necessary, I assume)

*We log transformed these variables before performing the paired t-test. In R, the procedure was performed as:*

*## Filtering pairs that were captured gap ##*

*pairs <- subset(gap, classe_RS == 1 & classe_BRK == 1 , select=c("Area_RS_m2", "Area_Brokaw_m2"))*

*## log-transform ##*

*log_area_RS <- log(pairs$Area_RS_m2)*

*log_area_BRK <- log(pairs$Area_Brokaw_m2)*

*log_dif <- log_area_RS - log_area_BRK*

*shapiro.test(log_dif)*

*## Paired t-test ##*

*t.test(log_area_BRK, log_area_RS, paired = TRUE, alternative = "two.sided")*

285: My takeaway from Table 2 would actually be that all distributions perform similarly (the dAIC is typically very low), and my guess would be that, if you account for the uncertainty of the small sample size, you cannot really differentiate between any of them here. I would highly recommend to test this! One interesting question is whether the field data have a slightly different exponent/shape, with a steeper decline at the largest gap areas (in line with the visual assessment). But, of course, sample sizes are very low.

*We thank Reviewer 1 for this valuable comment. We agree that the interpretation of calculated uncertainties is limited and will make this aspect clearer in the discussion. We will also discuss the small differences of the fits for the field and remote approaches. The power-law fit had the steeper slope among other functions and, in comparison with our field data, indicated a lower frequency of gaps greater than 100 m.*

315: I like this idea of calculating the biomass loss, and that at least one branch fall exceeded some of the uprooted/snapped tree losses. Could you put this into context of how much total biomass is stocked in the plot? I.e. what percentage is lost by gaps?

*We thank the Reviewer 1 for this suggestion. The losses of biomass in our studied gaps (1.35 Mg ha$^{-1}$ year$^{-1}$) account for 0.88% of the stocks in an old-growth forest contiguous to our plot (355.67 ± 34.53 Mg ha$^{-1}$ (mean ± standard deviation) (Amaral et al., 2019).*

334: This is not my favourite figure (and analysis). There are very few data points, and while I understand the general reasoning, it seems a bit like one could also pick a different percentile of extreme rainfall events, and the pattern might disappear. I suggest you remove this Figure and analysis.

*As previously mentioned in our answer to the comment 3, we suggest moving this figure to the Supplement. We will also improve the discussion on these results and include a couple of sentences exploring the limitations of our dataset for addressing the effects of the precipitation on gap dynamics.*

351-353: As noted above (and sorry for the repetition), there seems an inconsistency in the gap definition in the paper. If gaps are defined as openings in the upper canopy that clearly reach down to 2m (Brokaw), it does not make much sense to me to say that there are "no clear signs of opening in the upper canopy". Could it be that your field-based gap definition is slightly wider than the one you apply with the remote sensing/photogrammetry data, and is implicitly based around whether a tree has fallen? I am not saying that this is necessarily wrong, but that could explain divergences between both methods, because unless your photogrammetry approach overly smoothes the canopy, there is no a priori reason why it should not detect openings in the Brokaw sense, no? In this respect, I would also expect 2-3 sentences here on the problem of which of the two data sets (remote sensing or field) is the actual truth!

*We suggest editing the text as indicate below:*

*"Field data acquired by using the definition by* Brokaw et al. 1982 was considered as ground truth. Apart from simple and precise, these data allow comparing our findings with those from fundamental work conducted in other tropical forests. The Brokaw's definition has been used in ecological studies of gaps and to evaluate it with more recent remote-sensing tools (such as UAV photogrammetry) was one of the goals from our this work. *To accomplish with this goal, we used a confusion matrix for assessing the accuracy of our remote method of gap detection. Further, we calculated the percentiles of accuracy (a), precision (p), recall (r), and F1 Score (F) (Eqs. 1 − 4) (Dalagnol et al., 2021), where TP is true positive, TN is true negative, FP is false positive and FN is false negative:*

$$Accuracy\ (a) = ((TP+TN)\ /n)\ *100 \qquad (1)$$

$$Precision\ (p) = (TP/(TP+FP))\ *100 \qquad (2)$$

$$Recall\ (r) = (TP/(TP+FN))\ *100 \qquad (3)$$

$$Score\ F1\ (F) = (((2*p*r)\ /\ (p + r)))\ *100 \quad (4)$$

*The total number of correct detections is expressed as percentile. The p percentile indicates the ratio of positive predictions performed correctly based on all positive predictions (including false ones). The r percentile is used to access the ratio of correct positive-predictions in relation to all positive predictions. The F1 Score (F) is the harmonic mean between p and r, i.e., the mean between the errors of commission and omission; higher F-values indicate higher agreement between gaps identified in the imagery data and observed in the field (ground truth)."*

358: Unless I have missed it, I am not sure that the study shows how much gaps contribute to landscape patterns of biomass. It would help to put the losses into context of the whole-plot biomass stocks (cf. above), but I would still be wary of calling this "landscape" patterns. 18ha is probably not on the scale where landscape effects can be assessed, particularly, because power law-type distributions imply that you will have very few, very large gaps, and your plot may just accidentally miss out on extreme events / the long tails of the GSFD distribution (blowdowns/multiple emergent/canopy trees falling).

*As previously suggested, we will express rates of biomass losses in terms of area. We will also improve the explanation on the characteristics of our plot and its potential to integrate part of the landscape heterogeneity typical of our study region.*

362: again, what is an "understory gap"?

*We used this term for describing gaps restricted to the forest understory and that could not be in our UAV data. Meanwhile, our UAV data enable for the detection of height reductions of 10 m,*

*which allowed us to detect small gaps restricted to the higher portions of the canopy. We will edit for clarification.*

379: That the area of the gaps did not vary between methods is not entirely correct (cf. my comments above on this particular p-value), and even if we were to solely rely on the p-value, I would rephrase to say there was no evidence for strong variation in the area between the two methods (although in my opinion, there is some, limited evidence for divergences between the two methods in terms of gap area).

*We will improve the discussion on this topic and point out the limitations of our results.*

389: Cf. my comments before. I don't think, we can conclude that power-law distribution is the best distribution here, cf. also the large confidence interval of 2.137 +- 0.913! That is huge uncertainty!

*Thank you for the input regarding the uncertainty of the data. We will make this clear in the text and simplify the discussion regarding the distribution fits. Instead, we will focus on the differences between the remote and field detection.*

401: It seems to me that in many cases (not just your study), Weibull laws actually fit gap size frequency distributions better than power laws. I would discuss here what that would mean: it is more difficult to interpret (more parameters, not just one nice exponent), and it probably means that there is a change in generative mechanisms in gap formation across scales, which could make a lot of sense, because we probably shift from tree to branch level below a certain size threshold. You could also discuss this in the context of the typical tree size in your plot!

*Disturbances that are relatively small ($>5$ $m^2$) but generates height losses $>10$ m were detected. This includes branch fall or crown damage not resulting on individual mortality. We will improve the discussion regarding the size distribution of gaps and the resolution of our imagery.*

430-437: This motivation for rainfall patterns – correlation with extreme winds or lighting – should come much earlier in the paper (ideally in the introduction), so that the reader understands why these patterns are studied.

*As previously mentioned in the third comment by Reviewer 1, we will provide further information on the motivation of our study.*

448-449: I fully agree with your statement that forest inventories are fundamental, but I am not sure you showed conclusively that "mechanisms of formation could only be distinguished using field data". A very strong addition in my opinion (in Section 3.3.) would be to compare various attributes of the different gap types (branch/snapped dead/uprooted/standing dead), i.e. area/perimeter ratios, average height of lost canopy, average height within gap, canopy closure rate after gap formation (or even canopy changes before gap formation), and to really test whether there are indicators of how to differentiate different gap formation processes from remote sensing. My guess is, as you state, that you cannot reliably distinguish, for example, between uprooting and snapping, but it could still be that you would find a specific signature for standing dead trees (I imagine you could also use the RGB values of the orthophotos to identify them), or maybe you could find a

difference between branch falls and tree falls? This would be very interesting ecologically!

*We agree with this suggestion. However, this aspect is not within the scope of the present study. We are generating classifiers based on spectral, texture, and structural features of gap, and plan at presenting the related results in a next study.*

450-457: My impression is that the concluding paragraph is describing too many aspects at the same time (Weibull, mechanisms of gap formation, regional variation). Maybe focus on one priority, which seems to me the mechanisms/drivers of gap formation, and centre the paragraph around this notion?

*We thank the Reviewer 1 for this constructive comment. We propose to revise the Conclusions section as following:*

*By combining high temporal and spatial resolution UAV imagery with detailed field data on the mechanisms of gap formation, we could reliably assess the geometry and related losses of biomass for a closed-canopy Amazon forest. Although a larger proportion of canopy gaps could be detected from orthomosaics, their mechanisms of formation could only be distinguished using field data. Our results highlight that detailed forest inventories are fundamental for evaluating remote sensing products and metrics linking ecological processes to the carbon cycle. Future studies are needed to generate proxies for distinguishing mechanisms of gap formation and their landscape importance.*

**References**

[revised manuscript text omitted]

---

## Author Response (AR1)

**Author's response**

**Response to Reviewer 1**

The article "Gap geometry, seasonality and associated losses of biomass – combining UAV imagery and field data from a Central Amazon forest" studies gap formation on an 18ha field plot in the Amazon, using both remote sensing (photogrammetry/Structure from Motion) and field data. It provides an interesting look into canopy dynamics at one particular tropical forest site and a comparison (or validation) between field-based methods and remote sensing, which is crucial in linking traditional approaches with modern technology. Due to its substantial field sampling effort, the study can relate gap formation to different tree mortality modes and to associated biomass losses, thus linking ecological processes to the carbon cycle, which should be of great interest to readers of Biogeosciences. I also found the paper generally very well written, with well thought-through methods and clear and concise descriptions.

There are, however, a few changes/issues that I would recommend the authors to consider before publication. I will highlight a few larger aspects first, and then provide line-by-line comments in a classic review style.

*We thanked Reviewer 1 for their supportive revision and constructive comments. We stated that we would consider all aspects for preparing our revised manuscript. We also provided a point-by-point answer to all questions. The point-by-point answers have reference the lines of the revised manuscript.*

**1/ Definition of gap**: My impression is that the definition of gaps is not entirely consistent in the study. On the one hand, Brokaw's definition of gaps as extending down to 2m in canopy height seems to be used (l.161), but on the other hand, the authors argue several times that there are undetected "understory gaps", or gaps that are not visible in the upper canopy. Specifically, they attribute the differences between UAV imagery and field data to the UAV imagery not being able to detect such subtle changes below the canopy. But if we use the Brokaw definition, that should not be the case, as any gap would, by necessity, be a hole in the upper canopy and extend down to the ground, no? Could it be that the authors implicitly use treefall events or other canopy characteristics as part of their gap definition in their field-based studies? Could this also explain why gaps created by standing dead trees were the main difference? The definition aspect also affects what should be considered the "truth" for the validation – field-based assessments certainly offer more information to interpret gap formation (is it a branch fall or a tree fall? etc.), but to automatically consider them the truth (l.209) is not evident to me. Could one not argue that the 3D canopy height models derived from photogrammetry (or even better, lidar) can more accurately quantify height changes than visual/manual assessments?

*As suggested by Reviewer 1, we clarified the definitions of gap and revised the text for consistency.*

*We had used the definition by Brokaw (i.e., a gap in the forest canopy extending from the upper stratum to an average height of two meters above the ground) to compute and measure gaps in*

*the field (Brokaw, 1982). This was a classical and efficient method that allowed us to compare our findings with those from fundamental work conducted in other tropical forests.*

*Gaps had been identified through a detailed inspection of dead trees and fallen branches. For the identification of mortality events and description of mortality modes, we had followed the protocol of previous studies developed in our study region (Marra et al. 2014, 2018; Negron-Juarez et al. 2011; Ribeiro et al. 2016). To minimize errors, the inspections had been conducted across relatively narrow strips of forest (less than 10 m), and had always been led by ASLP. A field campaign had been undertaken to quantify and mark all pre-existing gaps (i.e., baseline).*

*Our remote-sensing approach provided detailed data on the upper canopy of the forest but no information on the understory. Gaps had been defined as disturbed patches with a total area > 5 m² and with reductions in canopy height greater than 10 m. These thresholds had been established based on the nominal resolution of our processed imagery (1 m) and the scale at which the forest inventories had been conducted, i.e., the tree level. Overall, the fall of branches and/or standing dead trees had produced severe damage mostly in the upper canopy, while the understory had remained intact. Therefore, upper canopy gaps detected remotely had not always been detected on the ground using the definition by Brokaw. This pattern showed that apart from covering relatively large areas at low costs, UAV photogrammetry had been an efficient method for monitoring gap dynamics - including the detection of upper canopy disturbance usually not visible from the forest ground.*

*We believed that our approach combining field with remote sensing data provided interesting insights on concepts and methods for quantifying gaps and their effects on forest dynamics. Classical methods based on field observations were efficient in detecting gaps extending from the upper canopy to the understory of the forest and were crucial for validating remote tools and for quantifying and modeling associated losses of biomass. High-resolution photogrammetry allowed for more precise measurements of the features of the gaps, including those restricted to the upper canopy and/or causing minor damage. Although more frequent, small-scale disturbances not implying tree mortality, such as defoliation and branch fall, were often neglected in forest inventories. Our study brought novel knowledge on the contribution of these events to ecosystem processes such as the carbon cycle. Quantifying the size-distribution of gaps and their landscape importance was crucial to understand how forests responded to shifts in the disturbance regimes triggered by climate change and land use.*

**2/ Study area size and gap size frequency distributions (GSFD):** Having such a detailed comparison between field based and remotely sensed gap structure is an important feat, based on substantial field work, so it makes sense that the authors focused on a plot size of "only" 18ha. However, this limits the analysis somewhat when it comes to assessing GSFD and the "landscape scale" patterns the authors are interested in. As expected for 18ha, sample sizes are very small (32 gaps in total, but only 14 gaps that co-occur in both field and remote sensing data). I am sceptical that such sample sizes yield much information on which distribution actually fits better, and I would expect the fitted Weibull, exponential and power law distributions to be so uncertain in their parameters (the power law exponent has an uncertainty of 2.137 +- 0.913, which is huge) that there is not much sense in comparing the fit of different GSFDs (one single data point might already shift the goodness of fit). If the authors would like to keep this analysis, I suggest they explicitly use confidence intervals / simulations of data generation to assess how reliably these distributions can actually be differentiated with so few gaps, or maybe focus less on which distribution fits better and more on the field-remote sensing comparison. They should also provide a careful discussion that does not place too much emphasis on

the different AIC values (which have generally low delta, anyways). More generally, if this type of analysis is carried out, I would also highly recommend the additional fitting of a lognormal distribution, which comes about through similar generative processes as power law models and is often an equally good fit.

*We thank the Reviewer 1 for this valuable comment. We used a bootstrap with 1000 interactions for calculating the confidence interval of the different fits and will include the information in the revised figure. We also tested lognormal as part of our analyses.*

*bsfitsizedistcontdata <-*
*function(sizedf,minsize=c(10,25),maxsize=c(720,900,NA),fitfcn=c("exp","pow","weib","logn"), nbootstraps=1000,alpha1=0.05, alpha2=0.01)*

*#  nbootstraps is the number of bootstrap interactions to run to get CIs*

*We had also used the Kolmogorov-Smirnov statistic to evaluate the goodness of fit (not only the AIC). We had also tested the lognormal distribution as part of our analyses. The lognormal distribution did not fit for values greater than 25 m2 in the UAV data, and the K-S statistic had also been adequate (those were the largest values for a good fit). As requested by the reviewer, we proposed to replace the following Figure S2 and Table S5.*

[Figure]

**Figure S2.** Size distribution of gaps at the INVENTA plot, Central Amazon, Brazil, for the period from September 18[th], 2018 to January 19[th], 2021. Size distribution of gaps detected with UAV photogrammetry (a) and field surveys (c). Modeled distribution of gaps larger than 10 m$^2$ detected with UAV imagery (b) and field data (d).

**Table S5.** Summary of fitting measures of the exponential, Power-law, Weibull and Lognormal functions describing the size distribution of gaps identified on the INVENTA plot, Central Amazon, Brazil.

| Detection method | Minimum size (m²) | Distribution | λ (95 % CI) | α (95 % CI) | K-S | Log likelihood | ΔAIC |
|---|---|---|---|---|---|---|---|
| UAV imagery | 10 | Exponential | 0.014 (0.008 – 0.031) | | 0.249 | -152.99 | 9.89 |
| | 10 | Power-law | 1.650 (1.529 – 1.831) | | 0.224 | -152.03 | 7.98 |
| | 10 | Weibull | 0.512 (0.266 – 1.460) | 21.544 (0.770 – 65.311) | 0.116 | -148.08 | 2.077 |
| | **10** | **Lognormal** | **3.559 (2.230 – 4.017)** | **1.105 (0.691 – 1.846)** | 0.107 | -147.04 | 0 |
| | 25 | Exponential | 0.013 (0.007 – 0.032) | | 0.2879 | -117.62 | 13.41204 |
| | **25** | **Power** | **2.094 (1.901 – 2.501)** | | **0.0760** | **-110.92** | **0** |
| | **25** | **Weibull** | **0.157 (0.078 – 1.685)** | **0.0002 (0.000 – 61.480)** | **0.0868** | **-110.75** | **1.661607** |
| | 25 | Lognormal | - | - | - | - | - |
| Field data | 9 | Exponential | 0.017 (0.009 – 0.032) | | 0.166 | -146.79 | 4.9063 |
| | 9 | Power-law | 1.614 (1.490 – 1.746) | | 0.266 | -153.01 | 17.3533 |
| | 9 | Weibull | 0.744 (0.477 – 2.186) | 40.370 (15.850 – 68.667) | 0.146 | -145.61 | 4.5635 |
| | **9** | **Lognormal** | **3.741 (3.376 – 4.070)** | **0.836 (0.506 – 1.265)** | **0.108** | **-143.33** | **0** |
| | 25 | Exponential | 0.017 (0.008 – 0.043) | | 0.215 | -111.134 | 4.5447 |
| | **25** | **Power-law** | **2.137 (1.811 – 2.670)** | | **0.158** | **-109.306** | **0.8869** |
| | **25** | **Weibull** | **0.414 (0.121 – 2.878)** | **4.712 (0.000 – 67.811)** | **0.144** | **-108.277** | **0.8305** |
| | 25 | Lognormal | 3.4735 ( -22.971 – 4.1895) | 0.975 (0.3404 – 3.7346) | 0.139 | -107.862 | 0 |

*To our knowledge, this is the first study merging field and remote sensing data for assessing the geometry of gaps and computing their contribution to processes regulating carbon cycle in Amazon. Our study region is covered with old-growth terra-firme forests including a topographic/edaphic gradient comprised of plateaus, slopes and valleys. Our 18-ha plot is monitored since the year 2000; all the trees >10 cm DBH are recorded and tagged (~10.500 individuals). Apart from relying on this long-term inventory, our study also used LiDAR and rainfall data available for the same plot. Flying a contiguous area allowed us to reduce costs and optimized the logistics of field campaigns, which was crucial for the success of a 28-months monitoring.*

*Our relatively large plot allowed for a robust quantification of tree mortality and associated losses of biomass across an environmental gradient. A total of 32 gaps were formed during the entire studied period; 18 were formed during a 14-month period for which field and remote data were acquired. However, our results should not be extrapolated beyond our study region. A regional assessment of the size distribution and geometry patterns of gaps would require the inclusion of further sites reflecting existing variations in forest attributes and disturbance regimes. We have improved the discussion on this aspect in the revised version of the manuscript.*

**3/ Precipitation and gap formation:** This part of the paper, while relevant, is not really motivated in the introduction, and more effort should be spent on explaining why it is relevant to suppose a link between precipitation and gap formation, and why presumably more direct drivers of gap formation (wind or even lightning) were not used. It is understandable that such data may not be available, but nothing in the introduction/methods section explains why precipitation is interesting. I would also remove the analysis of extreme rainfall events, because this seems like a filtering of the data that could be done with many thresholds (90th / 95th percentile, etc.), and with only 3 years and 8 data points for extreme rainfall (Figure 8), I doubt that the correlation the authors found tells us much about the system.

*We agreed with Reviewer 1 and planned to improve the text by explaining the links between precipitation and gap formation. In our study region, extreme wind gusts and precipitation had been identified as important mechanisms of tree damage and mortality (Chambers et al. 2013, Magnabosco Marra et al. 2018, Negrón-Juárez et al. 2018, 2023). A study that monitored tree mortality over five decades in a Central Amazon forest had found that trees died more frequently in wet months, even during drought years (Aleixo et al., 2019). Additionally, a positive correlation between precipitation and tree mortality had been reported for our study area (Fontes et al., 2018). A regional study based on 12 years of satellite data had revealed that major windthrows (visible on Landsat) in the Central Amazon occurred more frequently between September and February, months characterized by heavy rainfall, compared to the rest of the year (Negrón-Juárez et al., 2017). These background studies supported the expectation that a greater number of gaps could be observed during the rainy season. We revised the introduction to clarify this aspect.*

*We also understand that a comprehensive assessment of the influence of precipitation on gap formation requires longer-term data addressing seasonal and interannual variability. While this aspect could not be fully investigated with our 28-month dataset, it is part of an ongoing study which will incorporate a larger monitoring period and thus a higher number of extreme rainfall events and potential associated gaps. We aggregated background information on this aspect in the introduction and highlighted the limitations of our assessment. Furthermore, we had suggested moving Figure 8 to the Supplementary Material.*

**4/ Remote sensing vs. field data in assessing mechanisms of gap formation and biomass loss:** My impression was that section 3.3 would be one of the most interesting sections for readers of Biogeosciences, and that the authors could extend their analysis here a little bit without too much effort. For example, I would relate released biomass to overall plot biomass. There could also be an interesting comparison of released biomass visible from gaps, to overall biomass released from tree mortality, also counting understory mortality (if these data exist). Finally, since they have such a comprehensive data set, the authors could also compare other aspects of gaps between the different mortality modes (branch fall, snapped, etc.). I would suggest a look at the metrics the authors already calculated (gap geometry), but also previous and surrounding canopy height, and maybe also gap closure rates, with a focus on the values from remote sensing. Maybe, the authors could also use the RGB signature of the orthophotos as an additional metric to compare between mortality modes. Such an analysis would provide some hints on whether remote sensing/photogrammetry could distinguish different modes of mortality/gap formation/biomass losses, or at least separate one specific mode (standing dead). These are only suggestions and would, of course, only be indicative due to the small sample sizes, but I think they might be very interesting for future studies/Biogeosciences readers and be in line with the authors' objectives to assess how much we can learn from remote sensing compared to field-based assessments.

*Biomass estimations are indeed an important aspect of our study, and we thank the Reviewer 1 for stimulating this discussion. As suggested, we conducted calculations to compare the biomass released in gaps versus the stocks of old-growth forests in our study region. Additionally, we included a new figure in the Supplement (Fig. S3) to show the size distribution of gaps formed by branch fall and the different modes of tree mortality.*

[Figure]

**Figure S3.** *Size distribution of gaps formed by branch fall and different modes of tree mortality.*

*We agree that our RGB chronosequence can support research beyond the scope of this manuscript. For instance, we are investigating the geometric and reflectance signature of gaps created by branch fall and modes of tree mortality (i.e., standing dead, snapping and uprooting). This investigation may contribute to more reliable assessments of tree mortality events related to opposing climate extremes such as storms and droughts.*

*Our sample unit in this study is a gap event, which is not dependent on the size of the plot or subplot size. The18-ha plot was selected because (i) it is covered with an old-growth forest; (ii) includes a topographic/edaphic gradient (see answer to the second comment of Reviewer 2); (iii) has an infrastructure of trails and (iv) forest inventory, LiDAR and rainfall data.*

**Line-by-line comment:**

3: Is the title actually accurate? Gap geometry and seasonality do not seem to be such important results/aspects of this study, so maybe rethink/rephrase it?

*We agree and suggest the following title: "Gap attributes and associated losses of biomass - combining UAV imagery and field data on a Central Amazon forest".*

37: What is a multi-temporal process? Maybe rephrase?

*We suggest rephrasing as following: "The maintenance of these stocks depends on dynamic processes that regulate the growth and mortality of trees"*

41: Even though this may not be fully relevant to the paper, maybe droughts could be mentioned as another major extreme event?

*We agree that droughts are another extreme event that can influence gap dynamics by causing water stress and consequently increasing the frequency of branch fall and standing dead trees. We have included a comment on that in the revised version of the manuscript.*

50: This may be a definition question and not crucial, but in the context of tropical forests, gaps that are thousands of hectares in size (or tens of squarekilometers) seem unlikely, or probably not what tropical ecologists would commonly classify as gaps (e.g. one or several large canopy trees falling and leaving a gap in the canopy). Such a definition seems more common in fire-dominated boreal ecosystems. Maybe you could add one sentence specifically on tropical gap sizes. Also, this would be more in line with the extent of your sample plot.

*We thank Reviewer 1 for this important comment regarding the definition of gap size. This scenario of giant gaps is also common in the Amazon. Although with some surviving trees, gaps opened by extreme rain and wind in Amazonia can have a total area greater than 3,000 hectares (Nelson et al., 1994; Espírito-Santo et al., 2014; Negrón-Juárez et al., 2010, 2018, 2023). Recent evidences suggest hotspots for the occurrence of blowdowns (Negrón-Juárez et al., 2023; Urquiza Muñoz et al., 2021) and that the frequency of these events is regulated by atmospheric phenomena that are highly sensitive to climate change, such as the potential energy available for convection (CAPE) (Feng et al. 2023). We decided to include information about the influence of droughts on gap dynamics in our revised manuscript in lines 56 to 62.*

62-74: My impression is that this part of the paper jumps quite a lot between points, i.e. from the advantages of remote sensing, citing lidar remote sensing studies such as Dalagnol et al. 2021, to different definitions of gaps, to the problems of optical remote sensing. My question would be: Is the discussion of Landsat needed here, as UAV operates on a very different scale. A more interesting point might be how UAV photogrammetry differs from ALS/UAV lidar (e.g. no within-canopy structure,

no ground model, but likely cheaper, more flexible [although limited by meteorological conditions]).

*Our goal here was to discuss how UAV imagery could expand on previous assessments of windthrow tree-mortality, which up to date is mainly based on Landsat imagery (Nelson et al., 1994; Espírito-Santo et al., 2014; Negrón-Juárez et al., 2010, 2018, 2023). Although allowing for long-term studies at the regional scale, Landsat is only sensible for detecting relatively large gaps (>1,000 m²), which in Central Amazon usually involve the death of more than eight trees (Negrón-Juárez et al., 2011; Chambers et al., 2013).*

90: what are "traceable" modes of tree mortality? Or what would be "untraceable" ones?

*Recurrent field surveys following the routine and protocol established in previous studies (Magnabosco Marra et al., 2014, 2018; Ribeiro et al., 2016; Fontes et al., 2018) allowed us to clearly distinguish branch fall, standing dead, snapped and uprooted trees. For example, a standing dead tree may lose part or all of its crown shortly after dying. Thus, when field surveys are not conducted frequently, it is not possible to distinguish the mechanism of gap formation. A snapped tree is characterized by mechanical disruption of the stem (breakage or cracking). If checked in the field within a short period of time after its occurrence, it is still possible to find leaves attached the crown, thin wood/fibers at the snapping point/height and sap (e.g., resins and latex). Uprooted trees have exposed roots usually still attached to the trunk/crown. Branch fall gaps are formed when branches of living trees are broken and/or fall after dying. We have removed the "traceable" from the corresponding text.*

90: The last question with regard to rainfall is very adhoc and not really set up in the introduction. I would provide justification in the introduction on why precipitation should be relevant for gap formation. Would wind be a more important variable?

*Thunderstorms may propagate extreme wind gusts and rain. For instance, Araujo et al. (2021) found a significant relationship between gap formation and precipitation (extreme rainfall events, i.e. the rainfall rate in mm h$^{-1}$ of the 98.2th). We agree with Reviewer 1 and propose including more information to support why we expect precipitation to be an important aspect regulating gap dynamics and geometry. We include this information in lines 64 to 70.*

113-136: I am no expert in SfM/photogrammetry, but this seems well-described and a good workflow. I have one question: How did you deal with different meteorological conditions during planned flights (fog/rain)? Did you, for example, postpone scans during rainy days? Could this affect your results? How consistent was the timing of the acquisitions on average? I don't think this would be a major problem, but it would be good to mention this somewhere here.

*We thank the Reviewer 1 for this question. The flights were always carried around 09:00h and on the absence of fog and rain. The flights last approximately 15 minutes, which allowed us to acquire images under similar conditions of light. When possible, the flights were carried at cloudy conditions and diffuse light, which improves the visibility of the canopy while reducing shadow. We will add these details in the Supplement. We have added this information in the S1 text.*

155-172: This seems like a substantial effort and great, important work! Just out of interest: since you seem to have access to  EBA project's overlapping lidar data, is there a reason why you did not predelineate initial gap distributions from the lidar derived canopy height models?

*The LiDAR data was acquired in 2016. Although at a higher spatial resolution, these data could not be used as a reliable baseline for our study, which started approximately 2.5 years later.*

*Instead, we carried a detailed field survey to collect the coordinates and mark all existing gaps with a plastic-colored stick.*

197-209: This also makes a lot of sense. However, I would move the information from the last sentence (i.e. field value is considered true value) to the beginning to make it clearer to the readers what is considered the validation. I was wondering, however, whether in this case field data can actually be considered the true data? One could make a point that remote sensing (but maybe less so photogrammetry) actually provides a more accurate quantification of the 3D canopy canopy than visual/field-based assessments can. How would you justify your decision?

*We agree with Reviewer 1 and will carry the suggested change. The Brokaw definition was used to identify and describe gaps in the field. Apart from allowing comparison with previous studies, this method is compatible with our measurements of forest structure and losses of biomass. We also used the field data to validate the occurrence of gaps identified and measured remotely.*

215-223: While it is common to fit these distributions and the approach is methodologically sound, does this make sense here? 18ha is a very small area when it comes to gap delineation, so even without looking at the results, one would assume that your sample size is going to be so low that the inferred distributions are not telling us a lot (and the results bear this out, with 32 or 14 gaps in total). At the very least, I would expect simulations to construct confidence/credibility intervals that show how much variability there is and how uncertain the differences between the different distribution types are. My guess is that it would be very hard to come to any clear conclusion across 18ha. Also, would it not make sense to also test a lognormal distribution? The lognormal distribution is usually the one closest to the power law and comes about through very similar generative processes, so if you fit distributions.

*See answers related to the comment 2.*

225-229: Very interesting! I find the idea of quantifying released biomass very appealing.

*We appreciate this positive comment and agree that the field data is a highlight of our study.*

232: This process of calculating gap area formation rates sounds very complicated. Could you not just take the number/area of gaps that formed between each image acquisition and then divide the number/area by the time between each image? Assuming that images are taken at roughly the same intervals, that should give you a very sound estimate, no? Or am I missing something?

*Apart from relating number of gaps per unit of area, we aimed at analyzing the correlation between precipitation (measured as accumulated rainfall) and gap frequency, expressed as a percentage per month. We have improved the text for clarification.*

253: "which indicates that there was no traceable change in the upper canopy of the forest". This is probably more a discussion sentence anyways, but I find this problematic. According to the definition (Brokaw) you use, a gap is an "an opening in the forest canopy extending from the upper stratum to an average height of two meters above ground." So by definition something in the upper canopy has to change – either you don't pick it up in the photogrammetry data (maybe one of the processing algorithms is smoothing the canopy too much), or, alternatively, your

field-based assessment wrongly found a change in the upper canopy. This could also be an interesting question about gap definitions: should a standing dead tree already be classified as a gap, because light is reaching down almost without obstruction to 2m? How do you interpret this?

*We moved the text to the discussion. A gap easily identified in the field but not recognizable in the images could be one formed by the fall of lower branches from an emergent/canopy tree or the fall of a relatively small standing dead tree. These contribute relatively small area/volume. The respective changes could then be smoothed and gaps not detected. Since dense tropical forests have several vertical strata, the definition by Brokaw only fits when gaps extend to the upper canopy; this was not the case of relatively small damage promoted by branch fall. A standing dead tree was classified as gap when the light penetrated the understory of the forest leading to detectable changes in height/volume.*

260: I'm not sure the p-value is the best way to assess this here. Looking at Figure 3, one would guess that, at the large end of the gap spectrum, UAV seems to find larger gaps than field-based assessments (a difference of ca. 830 m2 to 580m2 for the largest gap seems substantial and larger than I would have expected). How did you derive the p-value? Did you log-transform the data beforehand (if you assume power-law/lognormal scaling, for example, that would be necessary, I assume)

*We log transformed these variables before performing the paired t-test. In R, the procedure was performed as:*

*## Filtering pairs that were captured gap ##*

*pairs <- subset(gap, classe_RS == 1 & classe_BRK == 1 , select=c("Area_RS_m2", "Area_Brokaw_m2"))*

*## log-transform ##*

*log_area_RS <- log(pairs$Area_RS_m2)*

*log_area_BRK <- log(pairs$Area_Brokaw_m2)*

*log_dif <- log_area_RS - log_area_BRK*

*shapiro.test(log_dif)*

*## Paired t-test ##*

*t.test(log_area_BRK, log_area_RS, paired = TRUE, alternative = "two.sided")*

285: My takeaway from Table 2 would actually be that all distributions perform similarly (the dAIC is typically very low), and my guess would be that, if you account for the uncertainty of the small sample size, you cannot really differentiate between any of them here. I would highly recommend to test this! One interesting question is whether the field data have a slightly different exponent/shape, with a steeper decline at the largest gap areas (in line with the visual assessment). But, of course, sample sizes are very low.

*We thank Reviewer 1 for this valuable comment. We agree that the interpretation of calculated uncertainties is limited and will make this aspect clearer in the discussion. We have moved the old Table 2 to Table S5 and included the lognormal function. We also discuss the discuss the small differences of the fits for the field and remote approaches.*

315: I like this idea of calculating the biomass loss, and that at least one branch fall exceeded some of the uprooted/snapped tree losses. Could you put this into context

of how much total biomass is stocked in the plot? I.e. what percentage is lost by gaps?

*We thank the Reviewer 1 for this suggestion. The losses of biomass in our studied gaps (1.35 Mg ha$^{-1}$ year$^{-1}$) account for 0.88% of the stocks in an old-growth forest contiguous to our plot (355.67 ± 34.53 Mg ha$^{-1}$ (mean ± standard deviation) (Amaral et al., 2019), (included in lines 319 to 321).*

334: This is not my favourite figure (and analysis). There are very few data points, and while I understand the general reasoning, it seems a bit like one could also pick a different percentile of extreme rainfall events, and the pattern might disappear. I suggest you remove this Figure and analysis.

*As previously mentioned in our answer to the comment 3, we suggest moving this figure to the Supplement (now Fig. S4). We also improved the discussion on these results and included a couple of sentences exploring the limitations of our dataset for addressing the effects of precipitation on gap dynamics (lines 458 - 460).*

351-353: As noted above (and sorry for the repetition), there seems an inconsistency in the gap definition in the paper. If gaps are defined as openings in the upper canopy that clearly reach down to 2m (Brokaw), it does not make much sense to me to say that there are "no clear signs of opening in the upper canopy". Could it be that your field-based gap definition is slightly wider than the one you apply with the remote sensing/photogrammetry data, and is implicitly based around whether a tree has fallen? I am not saying that this is necessarily wrong, but that could explain divergences between both methods, because unless your photogrammetry approach overly smoothes the canopy, there is no a priori reason why it should not detect openings in the Brokaw sense, no? In this respect, I would also expect 2-3 sentences here on the problem of which of the two data sets (remote sensing or field) is the actual truth!

*We suggest editing the text as indicate below:*

*"Field data acquired by using the definition by Brokaw et al. 1982 was considered as ground truth. Apart from simple and precise, these data allow comparing our findings with those from fundamental work conducted in other tropical forests. The Brokaw's definition has been used in ecological studies of gaps and to evaluate it with more recent remote-sensing tools (such as UAV photogrammetry) was one of the goals from our this work. To accomplish with this goal, we used a confusion matrix for assessing the accuracy of our remote method of gap detection. Further, we calculated the percentiles of accuracy (a), precision (p), recall (r), and F1 Score (F) (Eqs. 1 − 4) (Dalagnol et al., 2021), where TP is true positive, TN is true negative, FP is false positive and FN is false negative:*

$$Accuracy\ (a) = ((TP+TN)\ /n)\ *100 \qquad (1)$$

$$Precision\ (p) = (TP/(TP+FP))\ *100 \qquad (2)$$

$$Recall\ (r) = (TP/(TP+FN))\ *100 \qquad (3)$$

$$Score\ F1\ (F) = (((2*p*r)\ /\ (p + r)))\ *100 \quad (4)$$

*The total number of correct detections is expressed as percentile. The p percentile indicates the ratio of positive predictions performed correctly based on all positive predictions (including false ones). The r percentile is used to access the ratio of correct positive-predictions in relation to all positive predictions. The F1 Score (F) is the harmonic mean between p and r, i.e., the mean between the errors of commission and omission; higher F-values indicate higher agreement between gaps identified in the imagery data and observed in the field (ground truth)."*

358: Unless I have missed it, I am not sure that the study shows how much gaps contribute to landscape patterns of biomass. It would help to put the losses into context of the whole-plot biomass stocks (cf. above), but I would still be wary of calling this "landscape" patterns. 18ha is probably not on the scale where landscape effects can be assessed, particularly, because power law-type distributions imply that you will have very few, very large gaps, and your plot may just accidentally miss out on extreme events / the long tails of the GSFD distribution (blowdowns/multiple emergent/canopy trees falling).

*As previously suggested, we expressed rates of biomass losses in terms of area. Furthermore, we improved the explanation on the characteristics of our plot and its potential to integrate part of the landscape heterogeneity typical of our study region.*

362: again, what is an "understory gap"?

*We used this term for describing gaps restricted to the forest understory and that could not be in our UAV data. Meanwhile, our UAV data enable for the detection of height reductions of 10 m, which allowed us to detect small gaps restricted to the higher portions of the canopy. We have included further explanation of this in lines 360 to 368.*

379: That the area of the gaps did not vary between methods is not entirely correct (cf. my comments above on this particular p-value), and even if we were to solely rely on the p-value, I would rephrase to say there was no evidence for strong variation in the area between the two methods (although in my opinion, there is some, limited evidence for divergences between the two methods in terms of gap area).

*We changed line 387 to "4.2 There was no evidence for strong variation in the area between imagery and field data"*

389: Cf. my comments before. I don't think, we can conclude that power-law distribution is the best distribution here, cf. also the large confidence interval of 2.137 +- 0.913! That is huge uncertainty!

*Thank you for the input regarding the uncertainty of the data. We made this clear in the text and simplified the discussion regarding the distribution fits. Instead, we focused on the differences between remote and field detection. We include lines 412 to 417 and lines 424 to 431.*

401: It seems to me that in many cases (not just your study), Weibull laws actually fit gap size frequency distributions better than power laws. I would discuss here what that would mean: it is more difficult to interpret (more parameters, not just one nice exponent), and it probably means that there is a change in generative mechanisms in gap formation across scales, which could make a lot of sense, because we probably shift from tree to branch level below a certain size threshold. You could also discuss this in the context of the typical tree size in your plot!

*Disturbances that are relatively small (>5 m²) but generates height losses >10 m were detected. This includes branch fall or crown damage not resulting on individual mortality. We improved the discussion regarding the size distribution of gaps and the resolution of our imagery. This has also been included in lines 412 to 417 and lines 424 to 431.*

430-437: This motivation for rainfall patterns – correlation with extreme winds or lighting – should come much earlier in the paper (ideally in the introduction), so that the reader understands why these patterns are studied.

*As previously mentioned in the third comment by Reviewer 1, we provided further information on the motivation of our study (lines 72 to 79).*

448-449: I fully agree with your statement that forest inventories are fundamental, but I am not sure you showed conclusively that "mechanisms of formation could only be distinguished using field data". A very strong addition in my opinion (in Section 3.3.) would be to compare various attributes of the different gap types (branch/snapped dead/uprooted/standing dead), i.e. area/perimeter ratios, average height of lost canopy, average height within gap, canopy closure rate after gap formation (or even canopy changes before gap formation), and to really test whether there are indicators of how to differentiate different gap formation processes from remote sensing. My guess is, as you state, that you cannot reliably distinguish, for example, between uprooting and snapping, but it could still be that you would find a specific signature for standing dead trees (I imagine you could also use the RGB values of the orthophotos to identify them), or maybe you could find a difference between branch falls and tree falls? This would be very interesting ecologically!

*We agree with this suggestion. However, this aspect is not within the scope of the present study. We are generating classifiers based on spectral, texture, and structural features of gap, and plan at presenting the related results in a next study.*

450-457: My impression is that the concluding paragraph is describing too many aspects at the same time (Weibull, mechanisms of gap formation, regional variation). Maybe focus on one priority, which seems to me the mechanisms/drivers of gap formation, and centre the paragraph around this notion?

*We thank the Reviewer 1 for this constructive comment. We propose to revise the Conclusions section as following:*

*"By combining high temporal and spatial resolution UAV imagery with field data we could reliably assess landscape patterns of gaps and associated losses of biomass for a closed-canopy Amazon forest. Mechanisms of gap formation could only be distinguished in the field. Tree snapping was associated with the higher losses of biomass. Although with relatively lower losses of biomass, branchfall was the most frequent mechanism of gap formation. This finding highlights the importance of merging field and remote sensing data for assessing landscape processes regulating carbon cycle. Future studies could advance current knowledge by generating proxies for distinguishing mechanisms of gap formation using RGB UAV imagery."*

**Response to Reviewer 2**

The paper is very well written, it cites the proper literature and follows previous papers methodologies. Authors perform an extensive field campaign effort to collect the gap data, which is very laborious and I congratulate the authors. The study analysis relies on 18 gaps being detected during 2 years on a 600 by 300 m area from both field and UAV data combined for their analyses. From those gaps, a

total of 14 were detected by both approaches. My main concern is about the scale of the study and the sample size of gaps analyzed is too small (n=18?) in order to attack any of the four objectives proposed in the study. Thus, we cannot be sure if the results are indeed valid. See below two comments:

*We thank the Reviewer 2 for carefully revising our manuscript. **We considered all questions and comments while preparing our revised manuscript. Please find below a point-by-point answer to all the questions:***

*We believe there was a misunderstanding regarding the number of gaps used to address our research questions. The 18 gaps were only used to address the first question regarding the comparison of the two methods of detection (i.e., remote vs. field). To address the other questions, we used all gaps identified within the monitored period (32 in total). We edited the text for clarification and improved the discussion on the limitations of our study. As argued in our response to Reviewer 1, we also highlighted the importance of conducting similar assessments in regions with different forest structure and disturbance regimes.*

I don't fully agree with the first section title "Imagery and field data have different sensitivity for detecting small gaps". If you tell me a method finds 14 out of 18, that is, 77.77% accuracy, that is an excellent agreement and not exactly what the sub-section title suggests. However, the sample size is too small, if you detect or not an additional gap that changes by 5% the accuracy.

*We agree with Reviewer 2 and will change the title of this sub-section to "4.1 The mechanism of gap formation is related to the sensitivity of detection", focusing at the local level on the variation of sensitivity related to mortality modes. Indeed ~78% accuracy is an excellent agreement. The point we wanted to make is that the mismatch between the methods was mainly caused by relatively small gaps restricted to the upper canopy, which were consistently not detected in the field. As previously pointed out, we also improved the discussion on the limitations of our results.*

The scale analyzed does not allow to state that "UAV photogrammetry is a robust method for monitoring gap dynamics in Amazon forests". That may be a great step towards validating the detection of gaps in the field – which is a really challenging activity. However, the limitations of scale were not stressed out in Discussion. The method should be tested on larger scales before saying the method is robust. One suggestion would be to use airborne LiDAR as a reference for detection of the gaps, then using the UAV on top of it and this way gaining a lot of scale. Ducke forest nearby Manaus could be a great candidate for a future experiment. It already has a few airborne lidar flight lines in the past, and if more are collected, together with the UAV data, this could help a lot to couple with the UAV data.

*The goal of our study was to evaluate the accuracy of UAV imagery using field data collected simultaneously, to describe mechanisms of gap formation and to quantify associated losses of biomass. To our knowledge, such assessment has not yet been conducted for Amazon.*

*However, we agree that the limitations due to the scale of the study shall be better discussed. We included a discussion on the complementarity of UAV photogrammetry and LiDAR (lines 379 to 383). This complementarity stresses the potential of UAV photogrammetry, which has a relatively low cost, is simpler to process and thus can be repeated in other regions of interest. As the Reviewer 2 commented, future studies may expand the existing knowledge on the size distribution and dynamics of gaps by combining sensors with different resolution. When combined with LiDAR*

*and satellite, UAV imagery can be used to address patterns of gap formation and recovery at the landscape level.*

**References**

[revised manuscript text omitted]

---

## Author Response (AR2)

**Preprint bg-2022-251**

**Response to Reviewer 1**

The paper is an interesting and valuable contribution to the literature. Its unique value lies in connecting field-based measurements with remote sensing measurements, and then using this approach to link gap dynamics with biomass losses. As pointed out in previous reviews, the main underlying issue is one of sample size, but the authors have (mostly) addressed this now in their analysis and discussion. Similarly, earlier points on gap definitions precipitation have been addressed. I also highly value the authors' efforts in compiling and analysing the data as well as the detailed responses.

However, there are still two main issues that need to be addressed.

Interpretation/discussion of results: I found the interpretation of the results and the discussion sections difficult to follow and sometimes misleading. The clearest instance of a misleading claim was Discussion section 4.4, which states "Extreme rainfall-events control gap formation". This is a causal claim that is astonishing given both the study's setup (small spatial and temporal scales) and the study's results (at best weak correlations between gap patterns and precipitation). Similarly, section 4.3 states "Small-scale disturbances dominate canopy dynamics and associated biomass losses in Central Amazonia". This was also surprising, because the authors never correlate gap size and biomass losses (which they absolutely should, cf. below!). From the available materials, it seems to me rather that tree snapping, i.e., the mechanism responsible for the largest gaps, accounted for the largest biomass losses, so the opposite of what the paragraph claims. I may be mistaken, but then the authors need to show this. In contrast, sections 4.1 and 4.2, while not misleading, seem to summarize a wide variety of issues without a coherent storyline. Some of the points, such as detection sensitivity in the first paragraph need to be explained, other points should be summarized much more so that readers have clear takeaways. Part of my confusion may be due to the authors attributing properties of methods to the forest ecosystem itself. E.g., in stating "4.1 The mechanism of gap formation is related to the sensitivity of detection" they likely mean that the "classification of gap formation mechanisms" is related to detection sensitivity, not the actual mechanism. Another aspect may be that the authors tend to focus a lot on p-values/significance both in discussion/results, but not on effect sizes (e.g. how large are differences), and sometimes report with a precision that is not warranted for the small sample size (e.g. differences between 6, 7 or 8 gaps between mortality modes are not important, but seem more important when rendered as 2-digit precision percentages). I have made suggestions in the detailed comments to improve this.

*The last version of our manuscript already included clear statements on the limitation of our results due to the relatively low number of gaps. However, we would like to stress that our monitoring extended for more than two years and the sampling size also reflects the tree mortality rates of the forest ecosystem we have worked in. Still, we have revised the entire text again to address all points raised by reviewer 1 in the Discussion section of our revised manuscript.*

Biomass: I thank the authors for putting the overall biomass losses into the context of the carbon stocks at the site, but I still find that the analysis does not fulfill its potential here. I strongly

recommend adding one or two scatterplots/correlation plots to Figure 6. This would be one plot c) that plots biomass losses against gap size, and one plot d) that plots biomass losses against gap perimeter or GSCI. In each case, the data points should be coloured by gap formation mechanism. That would quickly tell readers how easy (or difficult) it is to infer biomass losses from gap formation, and would provide an interesting result + directly relate to Discussion Section 4.3.

*We thank the reviewer 1 for this suggestion and replaced Figure S3 by these scatter plots:*

[Figure]

**Figure S3:** Relationship between biomass loss and gap area (a) and between biomass loss and Gap Shape Complexity Index (GSCI) (b) at the INVENTA plot, Central Amazon, Brazil. The data were collected from September 18th 2018 to January 19th 2021. Gap area was determined from UAV Imagery data. X-axis in panel *a* is log-scaled.

Finally, I appreciate that it is hard to track all changes in revisions, but there are a quite a few paragraphs where words or descriptions are missing or where the authors have only responded to the reviewers, but not included the respective text in the article (e.g., log-transformation of variables, when performed, needs to be mentioned in the methods).

Detailed comments:

Title: "in a Central Amazon forest"?

*R: We corrected the word in the title.*

l.14: "In addition to detecting" could be removed, i.e., just write: "We measured the size …"

*R: The text was removed.*

l.17: "corresponding" instead of "associated"?

*R: We took the suggestion.*

l. 17: "either … or" not "either … and"

*R: The text was correct.*

l. 22: "Regardless of"

*R: The text was corrected.*

l. 23: "lognormal" should probably be lower case, as it is not a name (also in the rest of the text)

*R: We incorporated the suggestion throughout the text.*

l. 22-26: not sure whether I would put the different fits of lognormal/Weibull into the abstract. As stated in earlier reviews, given the small sample size such fitting exercises should not be over-emphasized, especially when there is no clear theoretical justification why one should be better than the other.

*R: We suggest that: "Regardless of the detection method, the size distribution was best described by a lognormal function for gaps starting from the smallest detected size (9 m2 and 10 m2 for field and imagery data, respectively), and the Weibull and Power functions for gaps larger than 25 m2. Properly assessing associated confidence intervals requires larger sample sizes."*

l. 26: I would reframe: "The main modes of tree mortality were not related to gap size, but to losses in biomass." The problem is, however: what does this have to do with gap patterns? Can we detect differences in mortality/biomass losses from the gaps themselves (through size/perimeter)? This is one of the main issues referred to above.

*R: We believe that it is possible to distinguish the mechanisms of gap formation by combining our UAV images and machine learning technics. However, this would require complementary (and extensive) fieldwork, and it was not within the scope of this study. We keep monitoring the plot and are currently working on these aspects, which will be addressed in an upcoming article.*

l. 27: Again, not sure whether this is such a strong result.

*R: We made clear that our results shall be carefully interpreted due to the relatively low number of gaps. Extrapolations of observed patterns to other Amazon regions also require validation of our method and further data sampling. However, as mentioned in the previous round, the significant association with extreme events aligns with what has been previously reported for the same region* (Negrón-Juárez et al., 2017; Fontes et al., 2018; Aleixo et al., 2019). *We have improved the text to make limitations clear and while suggesting issues that shall be addressed in further studies.*

l. 28: Does this not contradict l.26?

*R: Our goal here was to highlight the overall contribution of gaps. We worked again in the text to clarify that gaps ≥ 1 ha are not uncommon and as previously shown, can be traced using larger scale optical satellites (e.g., Sentinel-2, Landsat) (Marra et al. 2014; Negron-Juarez et al. 2011; Emmert et al. in review (doi: 10.20944/preprints202305.1631.v1).*

l. 32: This sentence is too harsh in my opinion. It's not like these results don't tell us anything about biomass/gap dynamics, it's just that we have to be careful about extrapolating. I would drop the sentence about extrapolation, or merge it with the following: "Future investigations combining remote sensing with field data are needed to confirm these relationships at landscape scale."

*R: The sentence was changed as following: "While combining remote sensing with field data has proven to be an accurate and precise method for mapping gaps compared to other existing approaches, it is important to note that our sample size is still relatively small. Therefore, the extrapolation of the results beyond our study region and landscape shall be made cautiously."*

l. 53: Maybe combine the sentences: "The size of gaps can vary … and defines the amount of light …"

*R: We combined the two sentences.*

l. 54: "Apart from related" – there is something missing

*R: The text was changed to: "In addition to being influenced by mechanisms of formation, the size and shape of gaps can also be influenced by local climate, extreme weather events, topography, soil, forest structure and species composition (Denslow, 1987; Marra et al. 2014; Araujo et al., 2021; Cushman et al., 2022)."*

l.58: Remove "Although with some surviving trees"

*R: The text was removed.*

l.57-70: I would merge these lines into one paragraph on wind and rain, and link them to the previous paragraph, e.g. by starting after "and related functions (Jucker, 2022)" with "In the Amazon, one of the key disturbances are extreme wind and rainfall events…." The following sentences need a bit of reworking/reordering.

*R: The paragraph was modified as following:*

*"Extreme wind and rain are major mechanisms of tree damage and mortality in Amazon forests (Nelson et al., 1994; Chambers et al., 2013; Magnabosco Marra et al., 2018; Urquiza Muñoz et al., 2021; Negrón-Juárez et al., 2023). Gaps opened by extreme wind and rain can have areas greater than 3,000 hectares (Nelson et al., 1994; Espírito-Santo et al., 2014; Negrón-Juárez et al., 2010, 2018, 2023). Recent studies have identified windthrow hotspots (Negrón-Juárez et al., 2023), with their frequency influenced by climate-sensitive atmospheric phenomena, such as convective potential energy (CAPE) (Feng et al., 2023). Previous studies reported higher frequency of gaps during wet months, even in drought years, which suggests a positive correlation between precipitation and tree mortality (Fontes et al., 2018; Aleixo et al., 2019). Additionally, satellite data support that large-scale windthrows (Magnabosco Marra et al., 2014b) visible on Landsat imagery (Negrón-Juárez et al., 2011) occurred more frequently between September and February, which are months marked by extreme rainfall (eg., > 30 mm hour-1; Negrón-Juárez et al., 2010, 2017)."*

l. 74-75: maybe make it a bit simpler "from small number of plots and infrequent surveys is a challenging task"

*R: The sentence was rewritten.*

l.76-77: I would focus only on optical methods here or rewrite this and the following with a clear separation of lidar/optical or satellite/airborne, otherwise it gets confusing for the reader. I.e., first you mention lidar, e.g. as in Greg Asner's studies, but then you move on to Landsat, which operates in a very different realm, both in terms of resolution and data type. Also, as an aside, while I agree

that airborne lidar generally provides high resolution and accuracy for e.g., canopy height, I would be more careful about this statement in the context of gap dynamics. Gap patterns depend a lot on CHM generation algorithms and the resulting surface roughness, so accuracy is probably not well-defined.

*R: We removed the sentences. The revised text starts with "In the Amazon, studies using intermediate spatial-resolution..."*

l.88: Again, it makes much more sense to only focus on optical data in the previous paragraph if the focus is on optical data here. Otherwise, a comparison with lidar would be needed. E.g., optical UAV are probably much cheaper to acquire and easier to handle, so practical (but they probably also come with a few disadvantages, such as not penetrating to the ground, being more sensitive to low clouds/mist).

*R: We changed the previous paragraph as suggested.*

l.114: This is not a crucial comment and I am not familiar with the INVENTA plot, but out of interest: this seems an extremely narrow height range for canopy trees (only ~50cm standard deviation in height?). From most height allometries at plot level that I know, tree height variation for trees > 50cm DBH is usually massive (several meters). Are canopy trees very narrowly defined here?

*R: This analysis was conducted with 267 canopy trees with DBH > 25 cm along a 1 km transect encompassing plateau, slope, and lowland areas. The analyses of tree proportion, diameter distribution, and growth rates between canopy and understory, as reported in* Araujo, 2019 *and* Araujo et al., 2020, *support the finding that the 267 trees analysed have reached the equilibrium phase of the successional stage and therefore have similar heights. For instance: (i) growth is not related to diameter for canopy trees and positively related for understory trees, consistent with the idea that light availability increases with diameter for understory trees but not for canopy trees; (ii) for canopy trees, the exponential distribution, typical of forests in equilibrium condition, is only fitted to trees with DBH > 25 cm; (iii) trees with DBH > 25 cm have more than a 50% probability of being in the canopy.*

l.198: could not find Magnabosco Marra et al. 2016 in references (was it 2014?)

*R: This reference was indeed missing in the previous version and is now included.*

l.230: "three" should be "four" now

*R: The suggestion was incorporated.*

l.245: "two-tailed" what? In your responses you mention the log-transformations, but this needs to be also part of the methods!

*R: We thank reviewer 1 for pointing this issue. We included the logarithmic transformations as part of the Methods section.*

l.268-289: This goes back to my questions about your gap definition in the previous round. It is important to differentiate here between what one of the methods registers (UAV) and what has happened in the actual forest (as seen, e.g., from all combined methods). Since the field measurements found three gaps that fulfill Brokaw's definition, it's not correct to say that "there was no traceable change in the upper canopy of the forest". There clearly was a traceable change, it was just not traceable by UAV. This should be carefully phrased throughout the article.

*R: We changed to: "These gaps, which were not visible on either the difference images or the orthomosaics, indicate an absence of detectable change in the upper canopy."*

l. 274-277: I would rephrase this (cf. also my comments from last time). A non-significant p-value is not the same as a small effect size (e.g., "no evidence for strong differences"). I would suggest: "UAV-inferred gap size exceeded the field-inferred gap size substantially for the largest gap (254 m2), and on average, the difference was 11.5 m2 (17% of mean field gap size), but non-significant (p = 0.85)."

*R: We changed the text as suggested.*

l.277-278: Make sure to check whether normal distributions actually are appropriate for those metrics as well, or if data were log-transformed or otherwise transformed, this needs to be part of the methods/tests descriptions. If possible, focus on effect sizes.

*R: The perimeter and shape complexity index (GSCI) data are log-transformed. We included this information in the descriptions of the Methods section.*

l.289: I do not understand what "The two most discrepant" refers to here

*R: These are the two lowest values of mean canopy height loss (Fig. 4). We changed the sentence to: "The two gaps with the lowest values of height loss (i.e., 1.13 m and 2.13 m) were only detected in the field (Fig. 4)."*

l.303-305: Please provide the number of gaps used for each analysis, and maybe use "best described" instead of "better described". As per the more general comments, it's fine to keep these results, but I would not put too much emphasis on them.

*R: We included the number of gaps used for each analysis and modified to the text as "best described".*

l.307-313: I really like this analysis, but I would not overemphasize the differences gap numbers. Differences between 8,7,6 (and even 11) gaps are hardly important. Also, I would not report percentages in Table 2, or at least not with 2-digit precision! With 32 gaps, a single gap results in a change of ca. 3%, so a precision < 1% is meaningless. You can always summarize as 34%, 25%, 22% and 19%, but probably 35%, 25%, 20% and 20% would be the most appropriate precision. I would write: "All mechanisms of gap formation accounted for a substantial share of the gaps created (from ~20% for standing dead trees to ~35% for branch falls). In contrast, contribution to total gap area was highly asymmetric, with ~60% accounted for by tree snapping, and <10% by standing dead trees.

*R: We rewrote the paragraph as suggested.*

l.317-321: As before, I would focus much less on the p-values. P-values usually tell us very little (for large sample sizes they are always < 0.05, for small samples, very rarely). How about effect sizes? Why not simply describe what you see in Figure 6a and 6b? From the graph it is clear that most gaps are very similar in size, irrespective of gap formation mode, but that snapping in this case has much more variance. So I would describe that. And from 6b, we see that all three tree fall modes have higher biomass loss than branch-fall, which also makes a lot of sense, so I would describe this! It would be important to add one or two panels showing the correlation between gap area (or gap perimeter) and biomass loss as a scatter plot. I.e., x axis is gap area/gap perimeter, y axis is biomass loss, and dots are coloured by mortality mode. I think what most people would be interested in is: if I measure the gap area and gap perimeter, how well can I predict the biomass loss?

*R: We reduced the emphasis on p-values and instead focused on the effects' size. We have also included the requested figure (please, see response to the overall comment).*

348: The header for this whole paragraph is a bit odd, because this point (sensitivity of detection, mechanism of gap formation) is barely explained and probably factually wrong. The mechanism of gap formation itself is likely not dependent on detection, because it is a property of the forest ecosystem

itself. I assume that the authors mean "The classification of gap formation methods is related to detection sensitivity". But again, this is also not explained, and the section as a whole lacks in coherence.

*R: The text was corrected as: "Detection is influenced by modes of tree mortality, branch fall and consequent gap features "*

348-350: By definition, the approach does not only underestimate the frequency smaller than the size threshold, it does not quantify it at all, so I would remove this.

*R: Removed.*

351=354: The caveat about needing more studies across different landscapes might make more sense later in the discussion, but not an urgent alteration.

*R: We sincerely thank you for the review. It is of utmost importance to emphasize the need for conducting studies across various landscapes. This aspect has been addressed throughout the manuscript to reinforce the accuracy of our data, avoiding any misleading interpretations.*

357-358: This needs to be explained more.

*R: We added to the text: "It is important to note that the sensitivity for detecting gaps was influenced by modes of tree mortality, which also lead to specific effects on species composition in regenerating patches of forest (Putz et al., 1983; Chao et al., 2009)."*

385: Again, the header is difficult to understand and does not summarize all the points that are discussed in the paragraph (GSCI, power law vs. lognormal). I would shorten these paragraphs a lot and clearly say what is essential. E.g., lines 425-430 seem to describe how the lognormal function is fitting the data, but it is very hard to follow. Also, as stated before, I would not focus too much on these results. I would just state the simple things: Lognormal was a bit better. This may reflect underlying processes, but should not be overinterpreted due to methods and small sample size. I don't think the results allow for much more inference.

*R: We changed the text to "Gap geometry and size structure: differences between imagery and field data". We also reduced the emphasis on model fitting and removed lines 425-430.*

431: This header also does not make much sense – is this really what your data show? A scatterplot of biomass loss vs. gap size would go a long way in explaining this, so I strongly recommend to make this clearer both in Results and Discussion! If I look at Figure 6b, some of the strongest biomass losses are recorded for tree snapping, whereas branch losses predictably have a lot of very small biomass losses.

*R: We changed the header to: "Ecosystem importance of mechanisms of gap formation and associated losses of tree biomass"*

431-448: The whole paragraph seems to summarize more the literature than relate to the results.

*R: We modified the text as suggested and included the following paragraphs for discussing our results:*

*"Repeated field measurements allowed us to quantify the relative importance of modes of tree mortality (i.e., standing dead, snapping and uprooting) and branch fall. Our results show that biomass losses did not differ among mechanisms of gap formation, but was relatively smaller in more frequent gaps formed by branch fall. Nonetheless, we could not distinguish different of gap formation based on the gap area and associated losses of biomass. These unclear differences indicate that quantifying the importance of varying mechanisms of gap formation (referring to the vectors that cause damage or death to trees and, consequently, gap formation) and modes of tree mortality (referring to mortality forms such as standing*

*dead, snapped, and uprooted trees) is a challenging task that require larger sample sizes and imagery data with higher spatial and spectral resolution.*

*Our findings corroborate those from a previous study conducted in Santarém (also in the Brazilian Amazon) using repeated high-density LiDAR data (Leitold et al., 2018). This study revealed that biomass losses due to single and multiple branch fall events accounted for only 20 % of the estimated biomass loss from canopy and understory trees. Similarly, in Panama, branch fall was associated with 43.5 % of gaps formed over a five-year period, but only for 23 % of the total disturbed area (Araujo et al., 2021b). The consistency of these findings across different tropical forests highlights the importance of tree mortality and canopy structure for regulating biomass stocks and balance.*

*To our knowledge, this is the first study combining remote gap-detection with direct measures of biomass (i.e., branches) and estimates based on a locally adjusted allometry. Almost half of the aboveground biomass of tropical forests (42 %, range of 12 % – 76 % across forests) is lost due to damage to live trees (Zuleta et al., 2023). If climate change results in a higher frequency of storms and extreme winds (Feng et al., 2023; Negrón-Juárez et al., 2023), branch fall and tree mortality rates can also be expected to increase. This may affect carbon stocks and balance, as well as the functional composition of these forests at the landscape level (Magnabosco Marra et al., 2018; Urquiza Muñoz et al., 2021).”*

449: The final header seems misleading. First, as outlined before, the sample size of the data makes this analysis problematic, but more importantly, if I understand the results correctly, there were no strong correlations in the data, so how do we come to the conclusion that “Extreme rainfall-events control gap formation”?

*R: We changed to: “Extreme wind and rainfall as potential mechanism of gap formation”.*

474-475: As stated in your abstract and in line with previous comments on sample size, the study cannot “reliably assess landscape patterns”. It can only assess local patterns and indicate what might hold elsewhere.

*R: Although at the local level, our study address landscape variations of topography and soil, including areas of plateau, slope and valleys. Since the scale of the study is clearly stated in the Abstract and the Material and Method section, we edited the text as following: “By combining high temporal and spatial resolution UAV imagery with detailed field data integrating landscape variations of topography and soil, our study provides novel and fundamental knowledge for understanding how tree mortality processes affect the structure and dynamics of Amazon forests.”*

475: “Mechanisms of gap formation could only be distinguished in the field.” To my knowledge this has not been shown in the study. If this is an important result, then it needs to be shown, e.g. by plotting not only gap area against gap formation mechanism, but all gap properties.

*R: This was supported by our data and as clearly stated in the text, refers exclusively to our study site and applied dataset. Nonetheless, we addressed the importance of this aspect in the Discussion section and pointed out that UAV images can be further explored to this goal. Other arguments complementary to this aspect are also provided in our answer to the comment in line #26.*

**Response to Reviewer 2**

This updated version of the manuscript is in much better shape and so I recommend the manuscript to be accepted as is. After reading the author's responses to my comments and reading the revised version, I got a better understanding of the number of gaps being mapped by either field and UAV.

The beginning of the Discussion highlights the excellent results for the match between field x UAV, then authors added a sentence acknowledging that more studies are required to further confirm these findings, because in the end we are taking an n=18 gaps with field + UAV to reach Conclusions. While the calculated F1-Score is excellent and trully above any other method that exists to map gaps, sample size is still very small to be sure this is a robust method. If I may still suggest something to be added, up to the authors, I think the manuscript would benefit (and I would be very much interested into reading about) the authors' perspective on a few sentences about potentials and limitations on scaling these maps of gaps to larger scales, that is, how to approach this in terms of data (sentinel-2, planetscope) and measuring gaps in the field for cal/val (recommendations for how to measure them in the field "quickly"), etc. Nevertheless, good job on the study and on undertaking the challenge to match field with remote sensing. I look forward to reading more manuscripts from this great team. Att, Ricardo Dalagnol.

*R: We appreciate the constructive and positive comments provided by reviewer 2. Following his suggestions, we included a detailed discussion on the limitations and opportunities of scaling up UAV maps to larger scales, such as satellites, and field measurements for calibration and validation of methods of gap detection:*

*"Scaling down is the first step to scale up processes and mechanisms regulating forest dynamics and functioning. This requires robust and validated remote sensing tools and method integrating regional variability of forest and environmental attributes. However, optical sensors with wide spatial coverage, relatively short revisiting time, and long data series are still limited in the Amazon. For example, in Landsat images, mortality events involving clusters with fewer than 6-8 trees cannot be identified (Negrón-Juárez et al., 2011; Chambers et al., 2013). Furthermore, the smallest gap size found in our study was 10 m² using a 1 m² elevation model and 2 cm resolution orthomosaics. The spatial resolution of Sentinel-2 is 10 m, while that of Planet is 3 m to 5 m. Therefore, the smallest gap size would correspond to just 1 pixel in Sentinel-2, potentially resulting in possible underestimation of small gaps. Extensive field inventory provides even more valuable information for scaling down (Fontes et al., 2018). However, it is necessary to develop pre-established field protocols that are reproducible and functional. These limitations can be mitigated by expanding drone-imaging coverage areas and increasing spectral information on targets, which should be the focus of future work. In conjunction with geometric patterns of gaps, as described in our study, higher spectral resolution can contribute not only to accurately distinguishing tree mortality modes and branch fall, but also to improving landscape estimates of biomass loss and recovery. Larger drones with the capacity to carry sensors designed to collect data across a broader spectrum range are fundamental for enhancing existing methods and establishing routines that enable detailed assessments of canopy dynamics and associated processes in dense and diverse tropical forests."*